



# Asymmetries in winter cloud microphysical properties ascribed to sea ice leads in the central Arctic

Pablo Saavedra Garfias[1], Heike Kalesse-Los[1], Luisa von Albedyll[2], Hannes Griesche[3], and Gunnar Spreen[4]

[1]Institute for Meteorology, Faculty of Physics and Geosciences, University of Leipzig, Leipzig, Germany
[2]Alfred Wegener Institute, Helmholtz Centre for Polar and Marine Research, Bremerhaven, Germany
[3]Leibniz Institute for Tropospheric Research (TROPOS), Leipzig, Germany
[4]Institute of Environmental Physics, University of Bremen, Bremen, Germany

**Correspondence:** Pablo Saavedra Garfias (pablo.saavedra@uni-leipzig.de)

**Abstract.** To investigate the influence of sea ice openings like leads on wintertime Arctic clouds, the air mass transport is exploited as humidity feeding mechanism which modifies cloud properties like total water content, cloud phase partitioning, cloud altitude, and thickness. Cloud microphysical properties in the Central Arctic are analyzed as a function of sea ice conditions during the Multidisciplinary drifting Observatory for the Study of Arctic Climate (MOSAiC) expedition in 2019-2020. A

state-of-the-art cloud classification algorithm is used to characterize the clouds based on observations by vertical pointing lidar, radar, microwave radiometer, and atmospheric thermodynamic state from the observatory on board the research vessel RV *Polarstern*. To link the sea ice conditions around the observational site with the cloud observations, the water vapor transport (WVT) being conveyed towards the RV *Polarstern* has been exploited as a mechanism to associate sea ice conditions upwind with the measured cloud properties. This novel methodology is used to classify the observed clouds as coupled or decoupled

to the WVT based on the location of the maximum vertical gradient of WVT height relative to the cloud-driven mixing layer extending above and below the cloud top and base, respectively. Only a conical sub-sector of sea ice concentration (SIC) and lead fraction (LF) centered at the RV *Polarstern* location and extending up to 50 km radius and azimuth angle governed by the time-dependent wind direction measured at the maximum WVT is related to the observed clouds. We found significant asymmetries for cases when the clouds are coupled or decoupled to the WVT, and when cases are selected by LF regimes. Liquid

water path of low level clouds is found to increase as a function of LF while ice water path does so only for deep precipitating systems. Clouds coupled to WVT are found to be low level clouds and are thicker than decoupled clouds. Thermodynamically, we found that for coupled cases the cloud top temperature is warmer and accompanied by a temperature inversion at cloud top, whereas the decoupled cases are found to closely be compliant with the moist adiabatic temperature lapse rate. The ice water fraction within the cloud layer has been found to present a noticeable asymmetry when comparing coupled versus decoupled

cases. This novel approach of coupling sea ice to cloud properties via the WVT mechanism unfolds a new tool to study Arctic surface-atmosphere processes. With this formulation long-term observations can be analyzed to enforce the statistical significance of the asymmetries. Our results serve as an opportunity to better understand the dynamic linkage between clouds and sea ice and to evaluate its representation in numerical climate models for the Arctic system.



## 1 Introduction

Cloud processes are among the major factors influencing the Arctic climate system. Compared to lower latitudes, Arctic clouds are more commonly occurring as mixed-phase clouds (MPC). MPC consist of ice crystals co-existing with supercooled liquid droplets and are predominantly located at low atmospheric levels (Mioche et al., 2015; Gierens et al., 2020; Korolev and Milbrandt, 2022). Because of their ubiquitous nature, MPC have a dominant role in important processes like precipitation and the surface radiative energy balance (Korolev and Milbrandt, 2022). The latter is particularly relevant during wintertime since it has been established that MPC have a significant impact in causing surface longwave radiative warming. This results in reducing the surface cooling rates thus being linked to the rapid Arctic warming (Serreze and Barry, 2011; Wendisch et al., 2017), which results in a wintertime Arctic amplification factor. In the central Arctic this factor is about 2.5 times higher than the current Earth's global warming signal (Wendisch et al., 2023).

There are still limitations on the understanding of the Arctic's persistent low-level MPC due to their counter-intuitive longevity despite instabilities arising from a variety of microphysical and dynamical processes. Surface-related interactions that foster turbulent and cloud-scale upward air motion are highlighted as important processes to maintain MPC under weak synoptic scale forcing (Morrison et al., 2012). Surface turbulence-driven heat and moisture exchange via updrafts can lead to relative humidity increases. These updrafts, when intense enough, can lead to situations of supersaturation with respect to liquid water, hampering the ice growing at the expense of liquid but instead fostering the simultaneous growth of ice particles and supercooled liquid droplets. When dynamic forcing is absent, MPC are generally unstable (Korolev and Milbrandt, 2022) and prone to ice growth at the expense of vapour deposition as expected from the Bergeron-Findeisen process (Bergeron, 1935; Findeisen, 1938). Feedback processes between the surface and clouds can foster the resilience of mixed-phase clouds when being dynamically coupled to the surface. In addition, local feedbacks among clouds, radiation, and turbulence together with moisture intrusions can lead to the persistence of MPC even in cases when the cloud is decoupled from the surface's energy and moisture sources (Morrison et al., 2012). Sources of surface energy and moisture in the Arctic are patches of open-water in the otherwise closed sea ice pack, such as polynyas or leads.

As defined by the World Meteorological Organization (WMO), leads are elongated areas of open water within the thick pack ice ranging from tens to hundreds of meters in width and tens to hundreds of kilometers in length. In the wintertime, leads are the natural sources of substantial heat and moisture flux thus warming the atmospheric boundary layer by transferring latent and sensible heat from the ocean to the atmosphere. In wintertime this process is governed by a large temperature difference between the air and water, and increases the atmospheric stability over ice floes (Lüpkes et al., 2008; Chechin et al., 2019), whereas in summertime the ocean and air temperatures are quite similar around $0\,°C$. Furthermore, as Lüpkes et al. (2008) concluded, when sea ice concentration is above 90% during winter, a change of 1% in sea ice concentration causes a temperature signal of +3.5 K on the near surface atmospheric temperature. Therefore leads provide an efficient mechanism to modify the atmospheric boundary layer to create unstably stratified conditions in contrast to the atmosphere





over the surrounding ice which is stratified stably (Andreas and Cash, 1999; Michaelis and Lüpkes, 2022). The extreme heat fluxes over leads are typically two orders of magnitude higher than over sea ice in winter (Andreas, 1980).

Recent studies based on the analysis of lead fraction of 200 km around the North Slope of Alaska, in Utqiaġvik, have shed light on the more complex interactions between leads and low-level clouds in the Arctic. Li et al. (2020a, b) found that although open leads foster the creation of low-level clouds; newly re-frozen leads tend to promote the dissipation of low-level clouds due to the cut-off of moisture while heat supply is still on-going. This counterintuitive result emphasizes the need to study the interaction of sea ice leads with clouds at smaller scales.

The Multidisciplinary drifting Observatory for the Study of Arctic Climate (MOSAiC) expedition from October 2019 to September 2020 was an international effort to study and characterize all aspects of the Arctic atmospheric-sea ice, ocean, ecology, and bio-geochemistry system in unprecedented detail, using a variety of approaches, and across multiple scales (Shupe et al., 2022; Nicolaus et al., 2022). MOSAiC is the most comprehensive measurement program conducted over the central Arctic. The obtained data provides the optimal framework to study coupled systems such as the interaction of sea ice leads and low-level clouds. This gives us the opportunity to scrutinize the effects induced by the occurrence of leads on low-level clouds and characterize the differences in cloud properties when leads are coupled or decoupled to the clouds.

The manuscript is structured as followed: In Sect. 2 the set of instrumentation used for this study on board of RV *Polarstern* is presented. Section 3 gives a detailed description of the methodology developed for the study and applied to a case study as an example is presented in Sect. 4, 4.1. The whole MOSAiC wintertime period is statistically analyzed and the statistical results for November 2019-April 2020 are presented in Sect. 4, 4.2. Conclusions and an outlook are given in Sect. 5. Supporting material for definitions, methodology, data processing, and further statistical results are summarized in the Appendix.

## 2 Instrumentation and data products

The suite of atmospheric remote sensing instrumentation on board of the RV *Polarstern* relevant for this study is mainly comprised of the Atmospheric Radiation Measurement (ARM) mobile facility AMF-1 of the US Department of Energy (www.arm.gov) and the OCEANET-Atmosphere container (hereafter refered to as OCEANET) of the Leibniz Institute for Tropospheric Research (TROPOS); (Engelmann et al., 2016). The list of instrumentation and data products utilized in this manuscript, along with their spatial and temporal resolutions, and references is summarized in Table 1. An extensive and detailed description of data availability of all MOSAiC instrumentation can be found in (Shupe et al., 2022, Table B1).

### 2.1 Ground-based atmospheric remote sensing

The primary set of ship-based remote sensing instruments for the observation and characterization of clouds are the ARM Ka-band zenith-pointing cloud radar (KAZR), a ceilometer (from ARM) as well as a PollyXT lidar and a microwave radiometer (MWR) HATPRO (both from TROPOS and installed in the OCEANET-container).

Figure 1 shows a typical synergy of observations by the KAZR cloud radar, the PollyXT lidar and the MWR for the case study of 18 November, 2019. These synergistic observations together with the atmospheric thermodynamic information provided by



weather models are imperative for the cloud type classification and retrieval algorithms for cloud macro- and micro-physics as
explained in Sect. 3.1.

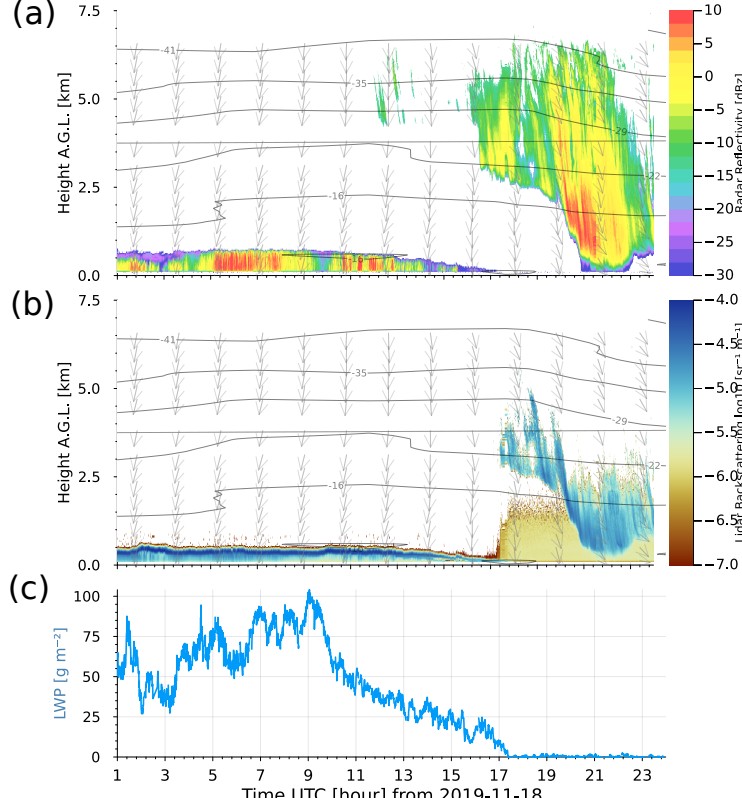

**Figure 1.** Synergistic observations with atmospheric remote-sensing instruments on board of the RV *Polarstern* from 18 November, 2019
during MOSAiC . (a) KAZR cloud radar reflectivity factor; (b) PollyXT lidar backscattering coefficient; (c) microwave radiometer liquid
water path (LWP). Shown isotherms (horizontally aligned) and wind vectors (vertical profile lines) were obtained from radiosonde data at
selected altitudes and time steps.

## 2.2 Radiosondes

For the characterization of the atmospheric thermodynamic state, the main information is obtained from radiosondes launched
from the RV *Polarstern* (Maturilli et al., 2021). For this study the high-resolution ARM Value Added Product (VAP) interpo-
lated sonde INTERPSONDE is being used. The INTERPSONDE is obtained from linear interpolation of the atmospheric state
variables from consecutive soundings into a fixed two-dimensional (2-D) time-height grid. The height and time resolutions are
20 m and 1 minute, respectively. The grid extends from 10 m up to 40 km altitude (Jensen et al., 2020). In order to account for
the thermodynamic interaction between the surface and the atmosphere, in this study the radiosonde vertical profile has been





merged with the Ground Infrared Thermometer (GND_IRT; Howie and Morris (2020)) as proxy for surface skin temperature $T_{\mathrm{gnd}}$ [K] which was assigned to altitude zero meters in the radiosonde profile.

Relevant atmospheric state variables needed for our methodology are provided or calculated from the radiosonde, e.g., pressure P [Pa], air temperature T [°C], specific and relative humidity $q_v$ [g g$^{-1}$], wind speed [m s$^{-1}$], and wind direction [degrees] (see Table 1 and references therein). Derived atmospheric state quantities are virtual potential temperature ($\theta_v$) [K], water vapour transport (WVT) [kg s$^{-1}$ m$^{-2}$], bulk Richardson number (Ri$_b$), planetary boundary layer height (PBLH) [m], and cloud-driven mixing layer height (CMLH) [m] above cloud top and below cloud base.

## 2.3    Satellite-based information for sea ice conditions


Space-borne sensors are the main source of information for long-term and large-scale monitoring of sea ice conditions in the Arctic. For this study, two main sea ice state variables are used: sea ice concentration (SIC), and lead fraction (LF).

### 2.3.1    Sea ice concentration

For the observation of sea ice concentration, satellite-borne instruments like the Advanced Scanning Microwave Radiometer
2 (AMSR2) and the Moderate Resolution Imaging Spectrometer (MODIS) are the most reliable instruments in terms of spatial and temporal continuity. These types of instruments, however, have limitations intrinsic to their measurement principles. AMSR2 is a microwave radiometer that is less influenced by clouds than optical sensors and has a good spatial coverage but is limited by its low spatial resolution of about 4 km at 89 GHz or coarser at lower frequencies. On the contrary, MODIS is an optical sensor and offers a higher spatial resolution of 1 km but its observations are restricted to cloudless conditions. In order

**Table 1.** Specifications of instrumentation and data products used in this study.

| Instrument/VAP | Full name | Variables/Producs $^\star$ | Resolution | | Reference |
|---|---|---|---|---|---|
| | | | time | range | |
| RV *Polarstern* central observatory | | | | | |
| HATPRO | passive microwave radiometer | LWP, IWV | 1 s | - | Ebell et al. (2022) |
| KAZR | Ka-band cloud radar | Ze, $V_D$, $S_w$, LDR | 3 s | 30 m | Johnson et al. (2020) |
| PollyXT | Multi-wavelength Raman Lidar | $\beta$, | 30 s | 7.5 m | Engelmann et al. (2016) |
| CEIL10m | Ceilometer 10m | $\beta$, CBH | 16 s | 10 m | Zhang et al. (2020) |
| INTERPSONDE | Interpolated Radiosonde | T, P, q$_v$, $V$ | 1 min | 20 m | Jensen et al. (2020) |
| GND_IRT | Ground Infrared Thermometer | $T_{gnd}$ | 1 min | - | Howie and Morris (2020) |
| Space-borne sensors | | | | | |
| MODIS-AMSR2 | $^\star$ | SIC | 1 day | 1.0 km | Ludwig et al. (2020) |
| Sentinel-1A SAR | $^\star$ | LF, DIV | 1 day | 700 m | von Albedyll and Hutter (2022) |

$^\star$ See appendix D3 for full definition of acronyms.





to exploit the best features of both sensors, Ludwig et al. (2020) have developed a merged 1 km MODIS-AMSR2 product by
tuning SIC from the MODIS 1 km resolution to preserve the AMSR2 average SIC.

The merged MODIS-AMSR2 sea ice product is of particular relevance for the present study since it provides the benefit of
potentially detecting open water leads within sea ice due to its finer resolution. We note that leads covered with thin ice, which
happens during winter conditions, are not necessarily detected. Nevertheless, for instance, the south-north aligned sea ice lead

on 15 April 2020 observed by Sentinel-1 SAR (Krumpen et al., 2021, Fig. 3), is resolved by the MODIS-AMSR2 SIC retrieval
(Ludwig et al. (2020)), but not by the 25 km resolution Ocean and Sea Ice Satellite Application Facility (OSI-SAF) product
(Lavergne et al., 2016) as shown in Fig. D2 (a) and D2 (b), respectively.

Nonetheless, a recent study by Rückert et al. (2023) has shown that warm air intrusion events occurring during the MO-
SAiC drift in April 2020, have fostered the formation of a large-scale surface glazing resulting in an underestimation of SIC

retrievals of about 30% which compromise the accuracy of the ASI algorithm used by the AMSR2 retrievals (Spreen et al.,
2008), thus affecting also the MODIS-AMSR2 product. Therefore, in order to evaluate the accuracy of the MODIS-AMSR2
product an alternative SIC product needs to be used. Here, the OSI-SAF SIC product (Lavergne et al., 2016) was chosen,
mainly because of its availability, coverage and higher accuracy during MOSAiC for April 2020 as shown by Rückert et al.
(2023). The details of the MODIS-AMSR2 SIC versus OSI-SAF product evaluation are described in Appendix C.

### 2.3.2  Divergence-derived sea ice lead fraction

The Satellites Sentinel-1 A/B from the European Space Agency use active microwave Synthetic Aperture Radar (SAR) in
C-band to capture the microwave properties of the sea ice. They are a valuable source to detect leads.

While the most common application is to classify leads from the backscatter coefficient of the SAR scenes (e.g. Murashkin
et al. (2018)), there is another approach that focuses on the formation process of the leads as seen in sea ice divergence (e.g.,

Kwok (2002), von Albedyll (2022)).

By calculating sea ice drift and sea ice divergence from sequential SAR scenes, leads show up in the sea ice divergence
whenever the ice moved apart from each other. Such lead fractions from SAR-derived sea ice divergence have the advantage
that they indicate the strong local change in ice velocity when a lead opens. They indicate the exact location of leads, are
independent of cloud coverage and their magnitude is directly linked to the widths of the leads without requiring sensor

calibration (Kwok, 2002).

Here, LF is calculated from divergence as described in von Albedyll et al. (2021) and von Albedyll (2022). The results
are interpreted as the average LF per grid cell which is subsequently drift-corrected and rendered with a spatial resolution of
700 m. One limitation on the lead detection by the divergence-based method is that only detects new openings. Stationary leads,
i.e. leads that do not open or close further, are not detected on the following days after the formation even though the leads

still exists. Those stationary leads during winter will likely be covered by thin ice. Figure 2 shows a comparison of MODIS-
AMSR2 and SAR SENTINEL-1 sensors for the sea ice situation on 18 November, 2019 around RV *Polarstern*, illustrating
the contrasting capabilities of MODIS-AMSR2 SIC (Fig. 2, a) and SENTINEL-1 divergence-based LF (Fig. 2, b) provide to
reveal in the sea ice.



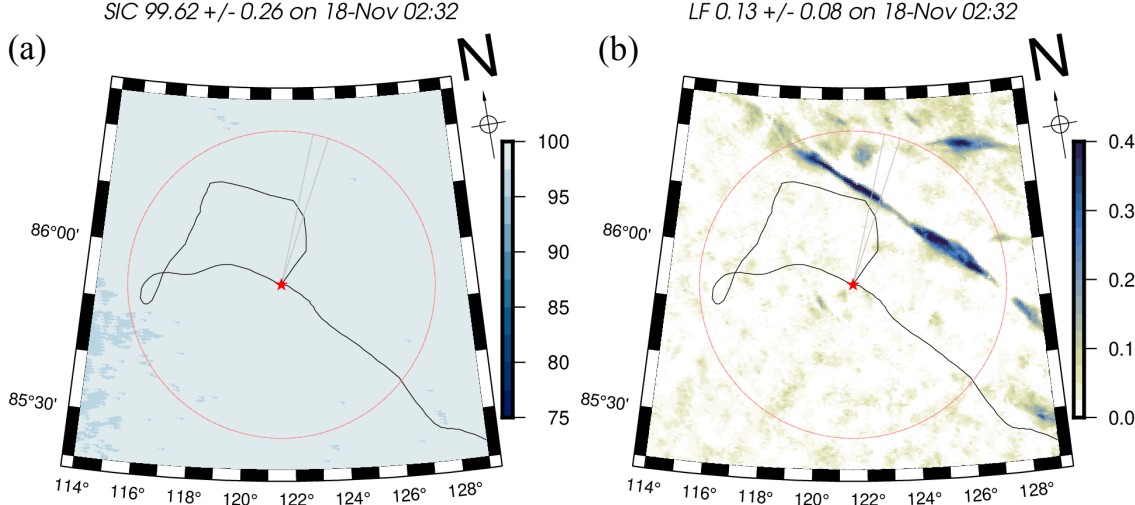

**Figure 2.** (a) SIC from MODIS-AMSR2 merged product from 18 November, 2019. (b) Lead fraction from SAR SENTINEL-1.The images are centered at the position of the RV *Polarstern* (red star) at the given date. The RV drift is indicated by the black line, the circle indicates the 50 km radius as region of interest. The grey cone indicates the relevant observation sector determined by the wind direction at the maximum water vapour transport (see text Sect. 3.4).

LF from SAR divergence-based data is available for the study period, except for the time between 14 January and 15 March
2020 (vertical dashed-grey lines in Fig. 3), when the RV *Polarstern* was north of the latitudinal coverage extending up to 87 °N of the Sentinel-1 satellite. To extract the mean LF of a certain region, e.g., 50 km around RV *Polarstern* (Fig. 3, top panel), the average of all grid cells that are located completely or partly in the region of interest are calculated.

## 3    Methodology

The present study focuses on the MOSAiC expedition from the 1st to 3rd leg which ranged from 11 October, 2019 to 16 May,
2020 thus covering the main part of the transpolar drift though the central Arctic (Nicolaus et al., 2022; Shupe et al., 2022).

To obtain the relevant metrics used to identify the cloud properties that can be associated with effects induced by the presence or absence of sea ice leads, two Arctic observables need to be linked, namely the clouds and the sea ice. This section details the methods applied for this purpose.

The conceptual model proposed to identify cases of sea ice leads with a potential to influence the cloud properties observed
aloft the RV *Polarstern* 's central observatory (CO) is depicted in Fig. 4 and described as following:

   – leads occurring spatially distributed within 50 km radius around RV *Polarstern* , are considered (Nicolaus et al., 2022),

   – leads release energy in form of heat and moisture to the atmosphere (Andreas, 1980; Michaelis and Lüpkes, 2022),





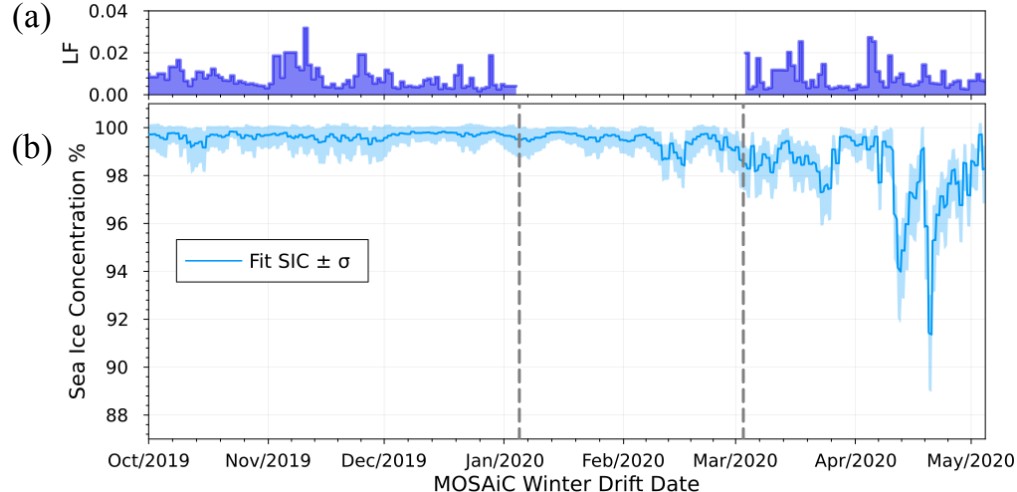

**Figure 3.** (a) Lead fraction (LF) estimated from SAR SENTINEL-1 divergence product within 50 km around RV *Polarstern* central observatory position, note the gap due to a lack of satellite overpasses at RV *Polarstern* latitude from 14 January 2019 to 13 March 2020; (b) Average fitted sea ice concentration (SIC) from MODIS-AMSR2 retrievals (blue). Shaded area corresponds to the SIC standard deviation of all pixels within 50 km.

– this release of energy can initiate a flux of water vapour along the wind direction or feed the already present horizontal water vapour transport (WVT) in the atmosphere,

– given the proper wind direction that WVT can move towards the RV *Polarstern* location,

– the WVT might favor the formation of new clouds or interact with already existing clouds by changing their properties,

– these clouds are then observed at the RV *Polarstern* CO.

According to this concept, the water vapour transport is an important component which serves as linking mechanism between sea ice leads and the clouds properties observed above the CO.


### 3.1 Cloud classification

The suite of remote sensing instruments, outlined in Sect. 2.1, observing the atmospheric state in zenith-pointing direction are the main source to estimate relevant cloud properties. Several procedures and algorithms have been developed to perform atmospheric target classifications based on synergistic ground-based remote sensing observations (e.g., Shupe (2007), Wang

et al. (2020)). Here we make use of the Cloudnet processing chain introduced by Illingworth et al. (2007). Cloudnet is an advanced classification algorithm specifically designed for the continuous evaluation of operational models using state-of-the-art ground-based instruments, Cloudnet not only provides the atmospheric target and cloud phase classification but also



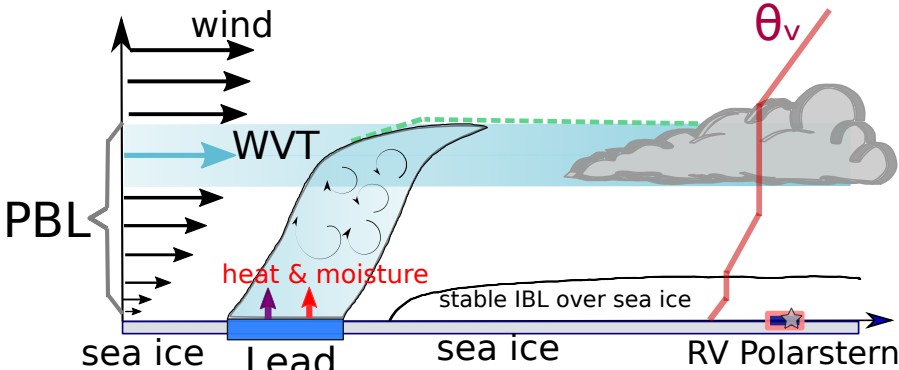

**Figure 4.** Conceptual model explaining how the presence of a sea ice lead can interact with the cloud observed above RV *Polarstern*. Water vapour transport (WVT) serves as conveyor belt for the latent and sensible heat released by the lead which can influence the cloud over the measurement site. PBL stands for planetary boundary layer and IBL for intermediate boundary layer.

an accurate prediction of the vertical and horizontal distribution of cloud micro-physical properties like ice and liquid water content (IWC, LWC; respectively) which are analyzed in Sect. 4. Basic requirements for the application of Cloudnet processing

routines are of having observations from a backscattering lidar, a microwave radiometer and a Doppler cloud radar. Cloudnet is continuously being improved and developed by a large community of users under an open-source scheme (https://cloudnet.fmi. fi) hosted by the Finish Meteorological Institute and coordinated by the ACTRIS initiative (https://www.actris.eu). Cloudnet has been selected as classification algorithm for this study in its open source version developed by the ACTRIS consortium (Tukiainen et al., 2020). Table 2 lists the input parameters and retrieval products obtained from Cloudnet, and Table 1 the

instrumentation used with Cloudnet in the present work.

Figure 5 (a) synthesizes an example of Cloudnet classification capabilities adapted to observations during MOSAiC. In addition to the target classification, The presented case study of 18 November, 2019 shows that the stratiform low-level cloud present until 17:00 UTC mostly consists of a mixture of supercooled liquid droplets and ice particles while the deep cloud system (present from 19:00 UTC onwards in Fig. 1) is mostly classified as ice-only.

A limitation of the Cloudnet classification and other synergistic retrievals happens in situations when the liquid cloud base is located below the lowest range gate defined as the height where all instruments are available, meaning the lidar signal is attenuated by low-level liquid clouds. This hampers the proper classification of liquid layers for which a lidar signal is required, hence all clouds are classified as pure ice clouds. This situation can be seen, for example, in Fig. 7 (b) from 16:00 to 17:00 UTC and Fig. D1 (b, from 14:30 to 18:00 UTC). For the MOSAiC wintertime period, about 29% of observed clouds were found

to have bases below the first radar range gate and identified as occurrence of low-level stratus according to the methodology developed by Griesche et al. (2020) based on information from the signal-to-noise-ratio of the PollyXT 532 nm near range channel.





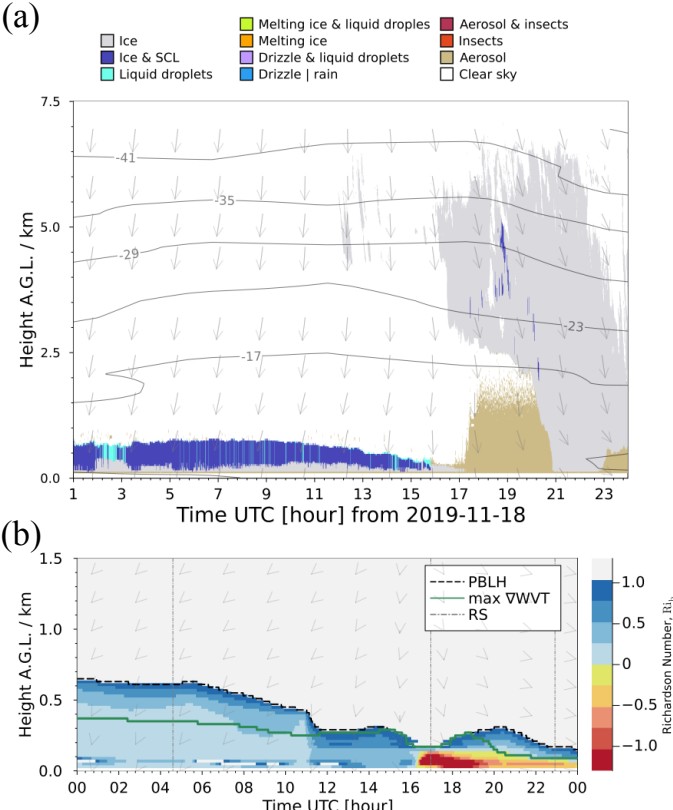

**Figure 5.** (a) Cloudnet classification for RV *Polarstern* observations on 18 Nov, 2019. Iso-therms and wind vectors are obtained from radiosonde profiles. Top colour coded boxes indicate the type of target classification. (b) Profile of the bulk Richardson number. The selected $Ri_c$ is visualized by black-dashed line as PBLH. The dark-green line indicates the location of maximum $\nabla$WVT within the PBLH. Radiosonde launch times are indicated as vertical dash-dotted lines (RS). Wind direction profile is indicated by arrows.

## 3.2 Atmospheric water vapour transport

The transport of air masses in the atmosphere is the main mechanism for the interaction of water vapour with the cloud
characterized as in Sect. 3.1. One widely-used concept to describe intense filament-like vapour transport in the atmosphere
is the one of Atmospheric Rivers (AR); (Martin-Ralph et al., 2020). Typically, either the vertically integrated -water vapour
(IWV) or -vapour transport (IVT) defined by Eq. B1 in Appendix B are analyzed to characterize ARs. Commonly AR are
identified whenever vapour flux exceeds a defined threshold relative to the zonal mean (Martin-Ralph et al., 2020; Zhu and
Newell, 1998). One disadvantage of both AR characteristic variables is that IVT - being an integrated quantity - does not carry
the information about the location of vapour transport in the vertical or whether one or multiple layers of vapour fluxes are
present at different altitudes. The same is true for the IWV, with an additional limitation being that IWV is a wind-independent
variable thus no transport is implicit with this metric.





| Product | Input parameter (Instrument) | Reference |
|---|---|---|
| Liquid water content | LWP (MWR) | Frisch et al. (1998) |
| | $Z_e$ (cloud radar) | |
| | T, p (radiosonde) | |
| | $\beta$ (lidar) [for liquid identification] | |
| Ice water content | $Z_e$ (cloud radar) | Hogan et al. (2006) |
| | T (radiosonde) | |
| Liquid droplet effective radius | $Z_e$ cloud radar | Frisch et al. (2002) |
| | $\beta$ (lidar) [for liquid identification] | |
| Ice crystal effective radius | $Z_e$ (cloud radar) | Griesche et al. (2020) |
| | T (radiosonde) | |

**Table 2.** Cloudnet product specifications and reference for retrieval methods.

For the present study, however, it is of main interest to monitor the transport of water vapour in the lower layers of the atmosphere where the interaction with sea ice or open ocean is mostly taking place. That is why a detailed analysis of the vertical changes of vapour transport in the lower atmosphere becomes of paramount interest to locate where the most relevant flux is located. To do so, we derive the vertical gradient of WVT starting from the standard definition of IVT (Martin-Ralph et al., 2020) given by Equation B1 detailed in Appendix B. The vertical gradient of WVT ($\nabla$WVT) is calculated using specific humidity $q_v$ [g g$^{-1}$], horizontal wind speed $\boldsymbol{v}_w$ [m s$^{-1}$], and air pressure $P$ [Pa] from radiosonde profiles with altitudes $z$ [m], following Eq. 1 whose detailed derivation is described in Appendix B:

$$\nabla\text{WVT} = -\frac{10^2}{g}\left|q_v \cdot \boldsymbol{v}_w\right|\frac{dP}{dz} \tag{1}$$

The advantage of using Eq. 1 is threefold: First, the altitudes at which local maximums of WVT occur can be identified separately from the flux $|q_v \cdot \boldsymbol{v}_w|$ profile; second, there is no need for thresholds to identify whether or not a layer of WVT is present (as is the case for IVT); and third, the derivative component $\frac{dP}{dz}$ behaves as a weighting factor (inverse exponential with altitude) that naturally gives more weight to fluxes at the lowest layers and diminishes the upper ones where meso-scale AR are more likely to be present. To constrain the relevant atmospheric layer even more, the $\nabla$WVT profile is analyzed only below the planetary boundary layer height (PBLH, estimation is explained in the following subsection). This allows to dismiss $\nabla$WVT peaks that are less likely to have interacted with sea ice in the vicinity of the RV *Polarstern*.

### 3.2.1 Estimation of the Planetary Boundary Layer Height

The Richardson number is defined as the ratio of turbulence associated with buoyancy to that associated with mechanical shear. This ratio resolved at increasing altitudes above surface level is known as the gradient Richardson number ($\text{Ri}$) and it is widely used to estimate the planetary boundary layer (PBL). However, when the atmospheric turbulence profile cannot be resolved with measurements at sufficiently high spatial and temporal resolution to resolve small scale turbulence - as in the case with





sounding the atmosphere - it is more convenient to use the bulk Richardson number ($Ri_b$) as a good indicator of the stability conditions in the atmosphere. The bulk Richardson number is defined as in Eq. 2:

$$Ri_b(z) = \frac{g}{\theta_v} \frac{\Delta\theta_v \, \Delta z}{(\Delta u)^2 + (\Delta v)^2} \tag{2}$$

where $g$ is the constant of gravity acceleration 9.81 [m s$^{-2}$], $\theta_v$ is the virtual potential temperature profile [K], $\Delta\theta_v = \theta_v - \theta_v(z_0)$, $\Delta u = u - u_0$, and $\Delta v = v - v_0$, the horizontal wind components [m s$^{-1}$]. The $\Delta z = z - z_0$ with $z$ the altitude of the atmosphere layers [m] and the subscript 0 indicates the surface reference.

The atmospheric stability is characterized by a range values of $Ri_b$, with $Ri_b < 0$ indicating an unstable and turbulent
atmosphere, $Ri_b < Ri_c$ neutral atmosphere and $Ri_b \geq Ri_c$ a stable with almost all turbulence diminished; with $Ri_c$ being the critical Richardson number. To estimate the PBLH by means of the $Ri_b$, the standard procedure relies on applying $Ri_c$ as threshold that defines the layer above which the atmosphere is considered to be non-turbulent and laminar. The most common $Ri_c$ used value is 0.25 although there is a wide range of values that can be found in the literature spreading from 0.1 to 1. For this study the $Ri_b$ has been calculated using the radiosonde data and the $T_{gnd}$ ground infrared thermometer (GND_TIR) as
surface skin temperature to estimate $\theta_v(z_0)$. For the purpose of this study an exact estimation for PBLH is not crucial but rather a rough approximation is useful to be used as atmospheric layer top to be considered for the determination of the altitude at which $\nabla$WVT reaches local maximum. The top of the PBL is then considered to be located at the altitude when the condition $Ri_b \geq 1$ is first met, with 1 being a rather conservative critical value to cover most of the relevant mixing layers for the Arctic winter atmosphere.

In Fig. 5 (b) the atmospheric stability based on the bulk Richardson number given by Eq. 2 and calculated form the ARM INTERPSONDE product is depicted for the Nov 18, 2019 case study. The atmospheric stability is colour-coded by light- to dark-blue colours ($0 < Ri_b < Ri_c$) for statically stable to neutral atmosphere. Above the critical values $Ri_c = 1$ (light-grey) the atmosphere is not considered to be significantly turbulent anymore, therefore the cloud within or above this level does not have potential to mix with the layers below. Unstable atmospheric conditions are highlighted by the yellow-to-red colours
corresponding to sub-zero $Ri_b$ which, for the case of 18 November, 2019, occur once the wind direction shifts to northerly and northwesterly directions and the deep cloud system is observed above RV *Polarstern*. For this case, the maximum of $\nabla$WVT within the PBLH varies between 0.1-0.4 km which is depicted by the solid green line in Fig. 5 (b).

In the following section, it is explained how $\nabla$WVT is being exploited as the mechanism responsible for the interaction between sea ice or open ocean and the clouds observations above RV *Polarstern*.

## 3.3 Cloud coupling

### 3.3.1 Cloud mixing layer

The cloud-driven mixing layer below the cloud base is determined by calculations based on the degree of variability of the virtual potential temperature $\theta_v$ profile (Eq. A1). A quasi-constant $\theta_v$ profile below cloud base height (CBH) implies a well-mixed layer. A departure from quasi-constant $\theta_v$ indicates a thermodynamic inversion, thus decoupling from the layer beneath.





To estimate the cloud mixing-layer height (CMLH) the relative variability of $\theta_v$ starting at cloud base downwards is analyzed. This concept has been extensively utilized by Sotiropoulou et al. (2014) and Gierens et al. (2020) for classification of surface coupled clouds. The criteria used in this study consists of calculating the cumulative variance of $\theta_v(i)$ ($\sigma_\Sigma^2(z)$ defined by equation A2) starting from cloud base towards the surface level or $i$=0. Thus CMLH is equal to $z$ at which the criteria $\sigma_\Sigma^2(z) \geq$ 0.01 K$^2$ is first met. The same procedure is applied to estimate the cloud-driven mixing layer above cloud top. The CMLH

below and above cloud-base and -top are quantities used to estimate the coupling or decoupling state of the cloud with the water vapour transport as described in the following section.

### 3.3.2    Cloud coupling classification

The observations are sorted into two classes depending on the likelihood of interaction between the sea ice situation downwind with the cloud observed aloft, and linked by the water vapour transport as conveying mechanism for moisture and sensible

heat to the clouds above the CO, i.e., whether or not the WVT is coupled or decoupled to the cloud. A cloud observation is considered to be coupled to WVT when the location of maximum $\nabla$WVT is found to meet one of the following criteria: be in the cloud, or between the cloud's CMLH below and above cloud-base and -top, respectively. Conversely it is considered decoupled when the maximum of $\nabla$WVT happens to be either above the cloud top's CMLH of below cloud base's CMLH. Those cases are illustrated in Fig. 6 for coupled (a and c), and decoupled (c) situations. It is important to note that in the present

study we are departing from the canonical concept of surface-cloud coupling generally found in the literature. This is due to the fact that the location of sea ice lead occurrence is not strictly co-located with the position of the RV *Polarstern* (see for instance Fig. 2 and D2), thus the sea ice-cloud interaction is not expected to be take place vertically within a static column, but rather dynamically ascribed by the air mass movement from afar. Moreover, the persistence presence of a surface temperature inversion in the Arctic makes the case of a vertically static column cloud-surface coupled a rare event.

Hence our definition of sea ice-cloud coupling status is governed by the following criteria:

**I. Coupled.**  when the maximum of $\nabla$WVT is localized within the cloud-driven mixed height above- and below cloud-top and -base, respectively (panel (a) and (b) in Fig. 6),

**II. Decoupled.**  when the maximum of $\nabla$WVT is found to be outside the cloud layer limited by the top- or bottom- CMLH, as shown in Fig. 6 panel (c).

According to the above, the classical definition of cloud-surface coupling is only a special case of this more general approach when the CMLH below cloud base reaches the surface level, i.e., $z = 0$ in Eq. A2. We found, however, that during the MOSAiC wintertime this situation only comprises of 4.7% of all cloudy observations and 7.3% of all cases that fulfill criterion **I. Coupled**.

### 3.4    Sea ice concentration in the direction of WVT

Information about the state of the sea ice is considered within a circular area of 50 km radius centered at the RV *Polarstern* central observatory (CO). This particular radius has been chosen as a compromise to cover the sea ice conditions





representative for the observations at CO based on SIC comparisons of circles with 6, 50 and 100 km radius (see Krumpen et al. (2021)).

Within the 50 km circular area, sea ice conditions relevant for the interaction with the cloud observations are extracted from
a conical sector centered at RV *Polarstern* and extended up to 50 km radius and angular span of 5 degrees. The azimuth angle of this conical sector is adjusted every minute based on the wind direction measured at the altitude of maximum $\nabla$WVT (green lines in Fig. 6 and Fig. 5 (b)). For instance, for the sea ice situation on Nov 18, 2019 only the LF and SIC highlighted within the grey lines in Fig. 2 is associated with the zenith-pointing cloud observations.

Although the drifting of the RV *Polarstern*'s ice floe (i.e., the geographical position for the center of the 50 km circular
area had an average drifting speed of 8.52 km per day (Krumpen et al., 2021; Nicolaus et al., 2022), we up-date the azimuth angle of the sea ice conical sectors every minute in synchrony to the available vertical wind profiles given by the ARM's INTERPSONDE product. Since LF and SIC information is only available in a daily basis, the center of the 50 km circular area also needs to be up-dated accordingly to avoid abrupt changes on the relative position of the leads respect to the RV *Polarstern*.

## 4 Results

### 4.1 Case Study of 18 November, 2019

Besides the Cloudnet retrieval products summarized in Table 2, further macrophyiscal and thermodynamical properties of the cloud are estimated from the Cloudnet target classification and from radiosonde. There properties include cloud base- and top

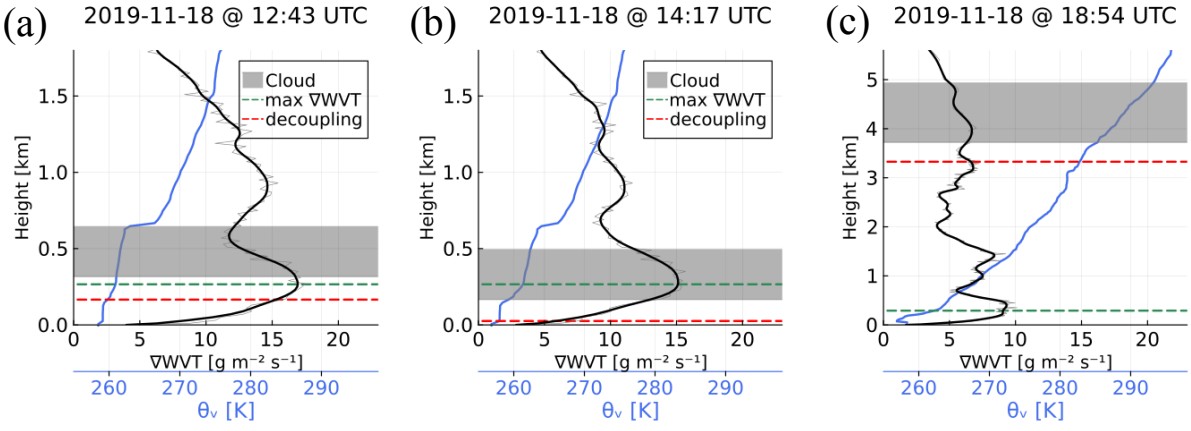

**Figure 6.** Examples for $\nabla$WVT profiles (black axes) coupling (a) and (b) and decoupling (c) to the cloud for 18 November, 2019. The liquid cloud layer is shown as grey shaded area. The coupling status is determined by the position of the maximum $\nabla$WVT (green-dashed) relative to the decoupling height CMLH (red-dashed): (a) the max $\nabla$WVT is above CMLH and below cloud base but still considered coupled to the cloud, (b) when max $\nabla$WVT is inside the cloud. Panel (c) shows a case when the cloud is decoupled. For reference the $\theta_v$ is indicated by the solid line and x-axis in blue.



height (CBH and CTH, respectively) as well as the temperature at cloud base and -top. Fig. 7 summarizes all those properties as well as the coupled/decoupled status of every observation according to Sect. 3.3 for the case study of 18 November, 2019.

Results presented in Fig. 5 (c) are based on vertical-integrals from single layer CBH to CTH of Cloudnet-determined LWC and IWC (see Table 2). LWP is determined for liquid-only clouds and MPC whereas IWP is determined for MPC and pure ice clouds and includes falling solid precipitation (snowfall).

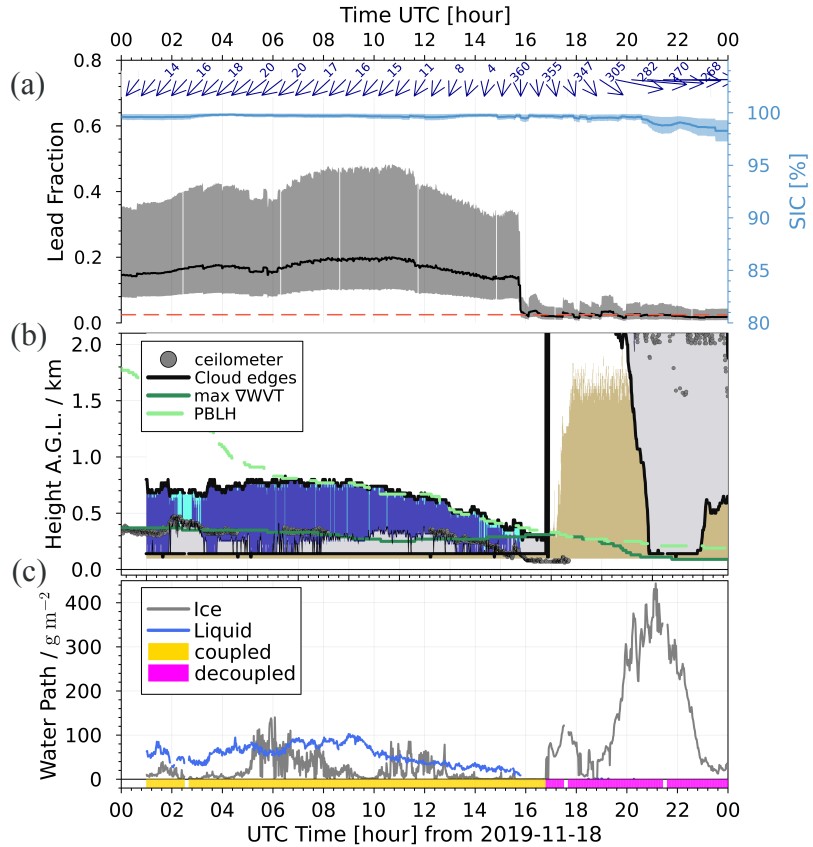

**Figure 7.** From top to bottom: (a) Median lead fraction time-series (black line, left axis) with the inter-quantile region (shaded region), and Sea ice concentration (light-blue, right axis) after considering only grid cells within a conical sector centered at RV *Polarstern*. The direction of conical sector as a function of the wind direction is shown by the blue arrows and values on top. The horizontal red-dashed line indicates an average LF value for totally covered sea ice. (b) same as Fig. 5 but magnified to the lower 2 km where the post-processing cloud edge detection is highlighted with black lines and the lidar cloud base is shown in grey dots. The height of maximum ∇WVT is shown in dark-green and the PBLH is in dashed light green. (c) IWP and LWP within the detected cloud layer (middle panel). The coupled/decoupled status flag is highlighted in yellow and magenta, respectively.

The case study in Fig. 7 (a) shows the corresponding 1-minute resolution time series for the sea ice statistics of all grid cells located within the conical sector aligned with the wind direction (see grey cone in Fig. 2) as described in Sect. 3.4. The conical





sector is adjusted as a function of the wind direction (blue arrows at top panel of Fig. 7) given at the height of maximum $\nabla$WVT occurrence (Fig. 7 b, green-solid line).

From 00:00 to 16:00 UTC of 18 November, 2019 latent and sensible heat were advected towards RV *Polarstern* from North-Northeasterly directions where a sea ice lead had formed (Fig. 2, right panel). LF within the conical sector has median values ranging from 0.1 - 0.2, and with inter quantile region (IQR) of up to 0.4 in Fig. 7 (top panel). During this time period of high

LF, a stratiform low-level MPC was observed (Fig. 1, 5 (a), and 7 (b) ).

At approximately 16:00 UTC, the wind direction changed towards the northwest to west, where no leads were located. This cut-off of the heat and moisture supply led to dissipation of the low-lever MPC. After about 17:00 UTC, a deep cloud system related to a storm was observed above RV *Polarstern*. This storm is associated with sublimation just above the maximum $\nabla$WVT and also led to an increase of turbulence in the lowest atmospheric layers (see Fig. 5 (b) red colours). SIC shows

a slight decrease from values near 100 % to about 98 % towards the end of the case study period. The coupling status is highlighted by the yellow (coupled) and magenta (decoupled) flag on the bottom of Fig. 7, panel (c).

From the obtained time series of extracted LF and cloud properties relationships between LF and micro- and macro-physical properties of the cloud are investigated. In order to reduce small scale variability in the following results (Fig. 8), for the case study of 18 November 2019, the observations are averaged in 10 minutes intervals, therefore every point represents $\sim$10

observations and the bars are the corresponding standard deviation within this time interval. Moreover, the results are sorted by cloud-WVT-coupling status with coupled cases represented by circles and decoupled cases by triangles.

From Fig. 8 (a) a clear relationship of LWP with LF can be seen, i.e. the larger the LF, the higher LWP, and the warmer the cloud top temperature. For IWP (Fig. 8 (b)), a less clear relationship is found, with a wide range of IWP values occurring independently of the magnitude of the observed LF. The only clear feature is the clustering of larger IWP at low LF which

correspond to the decoupled profiles of the deep cloud present after 17:00 UTC. Note that between 16:00 to 17:30 UTC (in Fig. 7 (b)) the lidar detects a liquid layer below the lowest available Cloudnet classification height, which Cloudnet could not relate this period with the occurrence of liquid droplets but instead it mis-classifies as ice cloud only. This is reflected in the total water path calculated within the lowest and top cloud limits as indicated in Fig. 7 (bottom panel) and shown as IWP $\sim 10^{-1}\,\mathrm{g\,m^{-2}}$ at the lower-left corner in Fig. 8 (b) below LF of 0.05.

Panel (c) in Fig. 8 depicts the liquid effective radius retrieved by Cloudnet and averaged over the cloud layer defined by

$$\bar{\mathrm{r}}_{\mathrm{eff}} = \frac{\int_{\mathrm{clb}}^{\mathrm{cth}} N(z)\,r_e(z)\,dz}{\int_{\mathrm{clb}}^{\mathrm{cth}} N(z)\,dz} \tag{3}$$

where $N(z)$ is the droplet number concentration, and $r_e(z)$ is the effective radius corresponding to the altitude $z$ within CBH and CTH. The best fit curve indicates a slightly positive correlation with LF and an increase of cloud layer thickness (exposed by the grey colours of the data points) as the LF increases.

Shown in Fig. 8 (d) is the in-cloud temperature lapse-rate defined as following:

$$\Gamma_{\mathrm{cloud}} = -\frac{dT}{dh} = -\left( \frac{T_{top} - T_{base}}{CTH - CBH} \right) \tag{4}$$



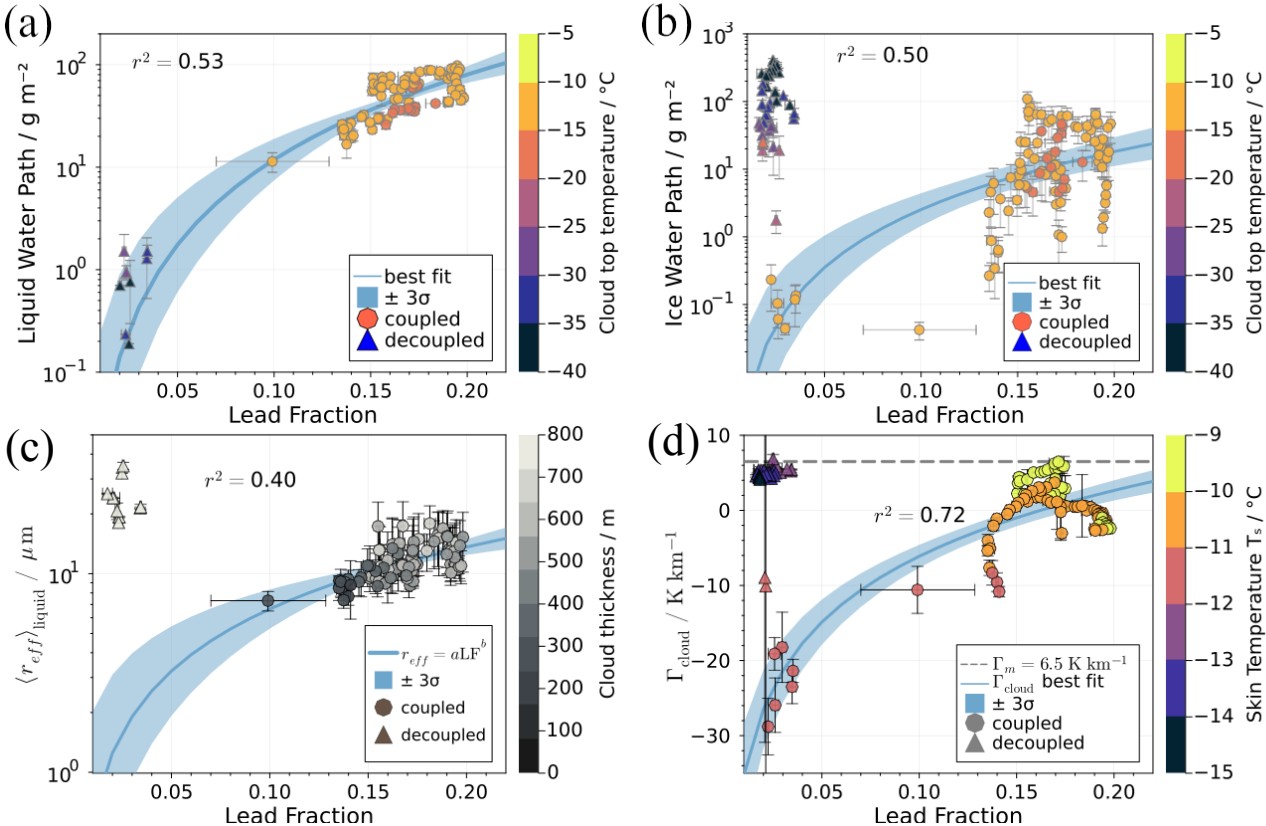

**Figure 8.** Results for 18 November 2019: (a) Mean single cloud layer LWP vs. LF (black-line in Fig. 7 top panel) colour-coded by cloud top temperature, coupled and decoupled cases are marked as circles and triangles, respectively. (b) Same but for IWP of same cloud layer. (c) Cloud layer average effective radius for liquid droplets, grey-colour-coded by cloud thickness. (d) shows $\Gamma_{\text{cloud}}$ as defined in Eq. 4 vs. LF with surface skin temperature as colour-coded, the horizontal dashed-line represents the moist adiabatic $\Gamma_m$. In all panels the best fit is shown as dark-blue-line and corresponding coefficient of determination $r^2$ based only on coupled data points.

Figure 8 (d) indicates that $\Gamma_{\text{cloud}}$ is often close to the moist adiabatic lapse-rate, whereas large negative values represent cases with a temperature inversion at cloud top. This feature is inconsistent with the data points between a LF of 0.13 and 0.14 where the $\Gamma_{\text{cloud}}$ shows a sharp drop from approximately $0\,^{\circ}\text{C\,km}^{-1}$ to $-10\,^{\circ}\text{C\,km}^{-1}$. The points are colour coded by the surface skin temperature, showing that coupled cases correspond to warmer surface as compared to the decoupled cases.

The presented 18 November, 2019 case study, nonetheless this case study encompasses a situation where the observed clouds have a high correlation with LF situation up-wind. To assess the robustness of the case study results over a wide range of cases, a statistics analysis is performed based on the same methodology applied to the whole wintertime MOSAiC expedition.






| Description | Number of Available Observations | | |
| | Total | Coupled [%] | Decoupled [%] |
|---|---|---|---|
| During wintertime | 259 200 | | |
| With cloudy sky | 199 926 | 121 970 [61%] | 77 955 [39%] |
| Cloudy and LF available | 124 787 | | |
| LF $\leq$ 0.02 | 104 025 | 66 432 [64%] | 37 593 [36%] |
| LF > 0.02 | 20 762 | 13 081 [63%] | 7 681 [37%] |
| Cloudy and SIC available | 199 333 | | |
| SIC > 98% | 108 979 | 69 332 [64%] | 39 647 [36%] |
| SIC $\leq$ 98% | 15 808 | 10 181 [64%] | 5 627 [36%] |
| Liquid clouds detected below first radar range gate | 68 998 [29%] | | |

**Table 3.** Number of available observations at 1 min resolution during MOSAiC wintertime for the statistical analysis. The number of data represents observations at a minute resolution. The total of observations during wintertime includes cloudless and cloudy observations. For cloudy situations, a further distinction is made between WVT coupled and decoupled.

## 4.2  Statistical Analysis

The methodology introduced for the case study in Sect. 4.1 was applied to the whole wintertime MOSAiC period (November 2019-April 2020). Table 3 summarizes the obtained dataset that was statistically analyzed after splitting for cases with LF less or equal to 0.02 (LF$\leq$0.02) and greater than 0.02 (LF>0.02). In that way we try to isolate cases in which sea ice leads have most likely been interacting with the observed clouds. Additionally the frequency of occurrences of different intervals of cloud water path sorted by the status of coupling to WVT for the whole period of analysis is presented in Fig. D3 (a) in Appendix D2.

Relevant differences are found in relation to the cloud micro-, macro-physic and thermodynamic properties between clouds classified as coupled or decoupled to the WVT.

Figure 9 depicts the probability distribution functions (PDF) of different macro- and microphysical cloud properties. The data is separated into four groups: WVT coupled cases (blue), decoupled cases (orange), cases with LF$\leq$0.02 (dashed-lines) and LF>0.02 (solid lines). In Fig. 9 (a) the PDFs of liquid layer base height is presented for coupled cases are generally comprised

of low level clouds with a probability of occurrence significantly enhanced for the subset corresponding to LF>0.02, with a main peak at 250 m. In contrast, decoupled clouds do not have a pronounced peak in liquid layer base height occurrence but are more homogeneously distributed over a range of a few hundred meters to one kilometer. Furthermore, Fig. 9 (b) exposes the fact that coupled clouds also tend to be thicker, with a peak in PDF at around 400 m, whereas decoupled clouds are equally likely to have thicknesses ranging between 40 to 500 m. Statistics of cloud top temperature are shown in Fig. 9 (c) where two

distinct features appear: clouds related to LF$\leq$0.02 have a maximum probability of cloud top temperature at around $-22\,°C$ regardless of their coupling state. Conversely, for LF>0.02, coupled clouds are generally warmer with maximum PDF of cloud



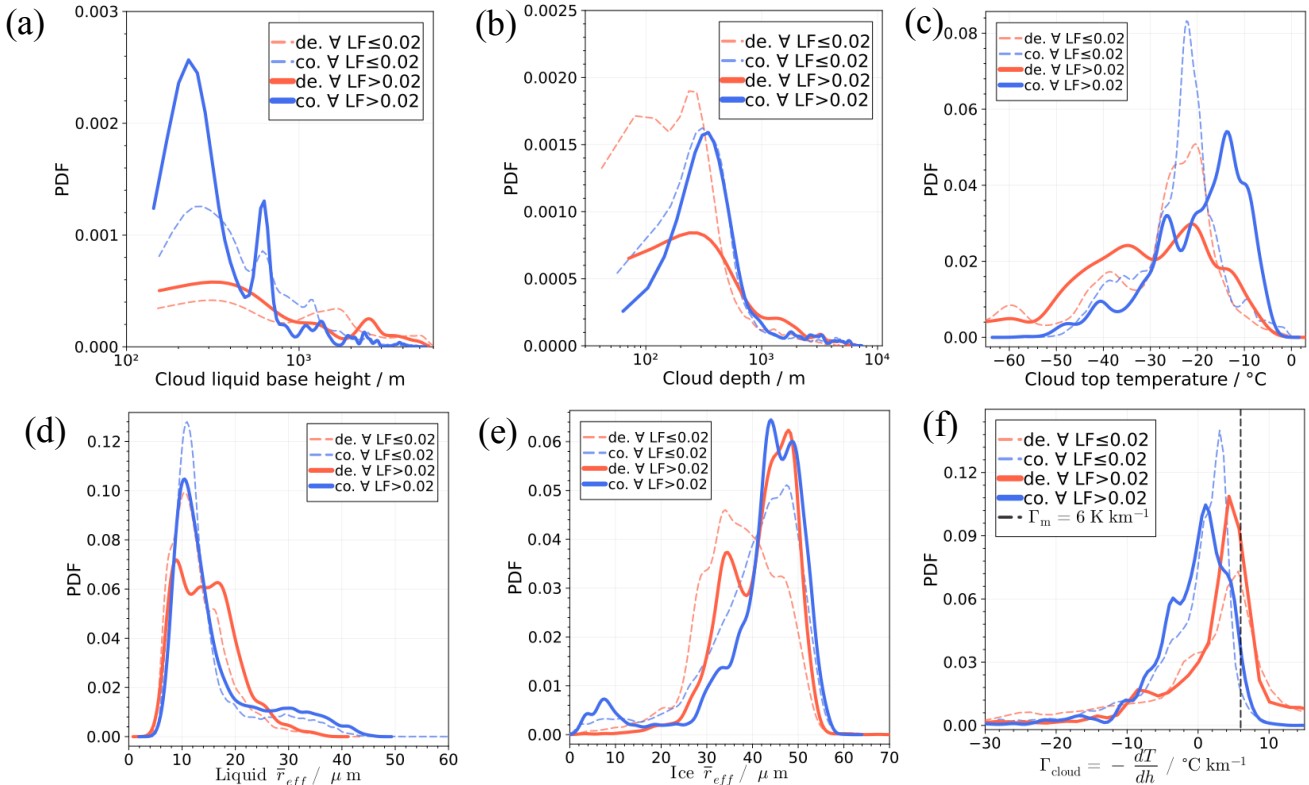

**Figure 9.** Probability distribution functions for all observations from November 2019 to April 2020. Data is sorted by LF≤0.02 (dashed-lines), LF>0.02 (solid-lines), coupled ("co.", blue), and decoupled ("de.", orange) cases of cloud properties for (a) liquid layer base height, separated for coupled (blue lines) and decoupled (orange lines) cases. Dashed-lines correspond to LF≤0.02 and the solid lines to LF>0.02. (b) Same but for the cloud depth. (c) PDF for cloud top temperature, (d) PDF for cloud layer mean effective radius of liquid droplets. (e) cloud layer mean effective radius of ice crystals, and (f) in-cloud temperature lapse rate, with the black dashed-line indicating a nominal moist adiabatic lapse rate.

top temperature at about $-12\,°C$ and a second minor PDF peak at $-29\,°C$. For decoupled clouds and LF>0.02, the cloud top temperature PDF spreads out to colder temperatures with a primary peak at $-22\,°C$ and a second peak at $-36\,°C$. Moreover cloud top temperatures below $-40\,°C$ are considerably more frequent to be observed for decoupled than coupled clouds.

Figure 9 (d) the PDF of cloud layer mean liquid droplet effective radius $\bar{r}_{\mathrm{eff}}$ of coupled cases peaks at 12 $\mu$m for both LF classes. The PDF of $\bar{r}_{\mathrm{eff}}$ for the WVT-decoupled cases exhibits a bimodal distribution peaking at 8 $\mu m$ and 17 $\mu m$ irrespective of LF. Likewise, the average effective radius for ice particles (Fig. 9 (e)) exposes a bimodal PDF of ice $\bar{r}_{\mathrm{eff}}$ with peaks at 42 and 49 $\mu$m almost equally likely to occur for coupled cases, while the PDF for decoupled cases have one minor peak at 32 $\mu$m and a major peak at 48 $\mu$m. The PDFs for LF≤0.02 (dashed-lines) have single maximums at 48 and 32 $\mu$m for coupled and

decoupled cases, respectively.





To complement the thermodynamic features of the cloud, Fig. 9 (f) shows the PDF of cloud layer temperature lapse rate. The main feature found is that for LF≤0.02 the decoupled PDF indicated a maximum at the nominal value for moist adiabatic lapse rate $\Gamma_m$, while decoupled clouds at LF>0.02 show a slightly lower lapse rate.

The coupled $\Gamma_{\mathrm{cloud}}$ PDFs are biased and skewed towards negative lapse rates with the most probable value found to be 385    0 °C km$^{-1}$ and -2 °C km$^{-1}$ for LF≤0.02 and LF>0.02, respectively. The latter also has a minor peak in the PDF around -5 °C km$^{-1}$. The dominant feature found in Fig. 9 (f) is that the $\Gamma_{\mathrm{cloud}}$ PDFs of clouds coupled to the WVT mechanism are displaced towards lower or even negative $\Gamma_{\mathrm{cloud}}$ (i.e., temperature inversion above the cloud base) and this characteristic is enhanced when LF>0.02.

### 4.2.1 Fraction of ice water content in cloud

Hereafter we define the fraction of ice water content in the clouds relative to the total condensed water in the cloud (ice and liquid) as follow:

$$\chi_{ice} = \frac{IWP}{IWP + LWP} \tag{5}$$

The definition given in Eq. 5 is in line with Korolev and Milbrandt (2022) phase composition of clouds but differs from most of the studies based on space-borne Arctic observation (Coopman et al., 2018) where the $\chi_{ice}$ is defined as the number of 395    grid cells considered to be ice divided by the number of grid cells considered as either liquid or ice. Similarly, the definition in Eq. 5 differs from other ground-based and ship-based observations over mid-latitudes and the Arctic where the fraction of ice containing clouds with respect to all observed clouds is considered (Kanitz et al., 2011; Westbrook and Illingworth, 2011; Griesche et al., 2021) regardless of the water content in those clouds. Therefore the results based on ice water fraction analysis presented here cannot be directly related to the previously mentioned work. We mainly are interested in the features controlling 400    the cloud microphysical properties such as LWP and IWP since those are the dominant drivers for the cloud-surface interaction. Furthermore, note that in the following analysis Eq. 5 is only applied to the single cloud layer relevant for the classification of coupled or decoupled to the WVT, therefore the results are not representative of the whole atmosphere in case of multi-layer cloud situations.

Fig. 10 depicts a clear difference of ice water fraction when separated by the cloud coupling to the WVT status (blue for 405    coupled and orange for decoupled) between -15 and -25°C for cases with LF≤0.02 (a) and between $-12$ and $-30\,°\mathrm{C}$ for cases with LF>0.02 (b). For the situation in (a) the ice water fraction, for coupled and decoupled cases, increases until a local maximum corresponding to a cloud top temperature of $-15\,°\mathrm{C}$ as indicated by the vertical light-blue dashed-line. This can be explained by the fact that at approximately $-15\,°\mathrm{C}$ the maximum ice growth takes place due to the largest difference between saturation water vapour pressure over ice and water (Rogers and Yau, 1991). Below $-15\,°\mathrm{C}$ cloud top temperature the coupled 410    and decoupled cases depart significantly until approximately $-25\,°\mathrm{C}$, with the coupled cases showing a steady increase of ice water fraction presumably due to the intake of humidity provided by the coupling with the WVT, thus fostering the formation of ice particles, whereas the decoupled case indicates a drop of ice water content up to $\chi_{ice}$=25% at a cloud top temperature of $-20\,°\mathrm{C}$ whereafter the heterogeneous freezing process continues. Both $\chi_{ice}$ curves reach 50% at about the same temperature





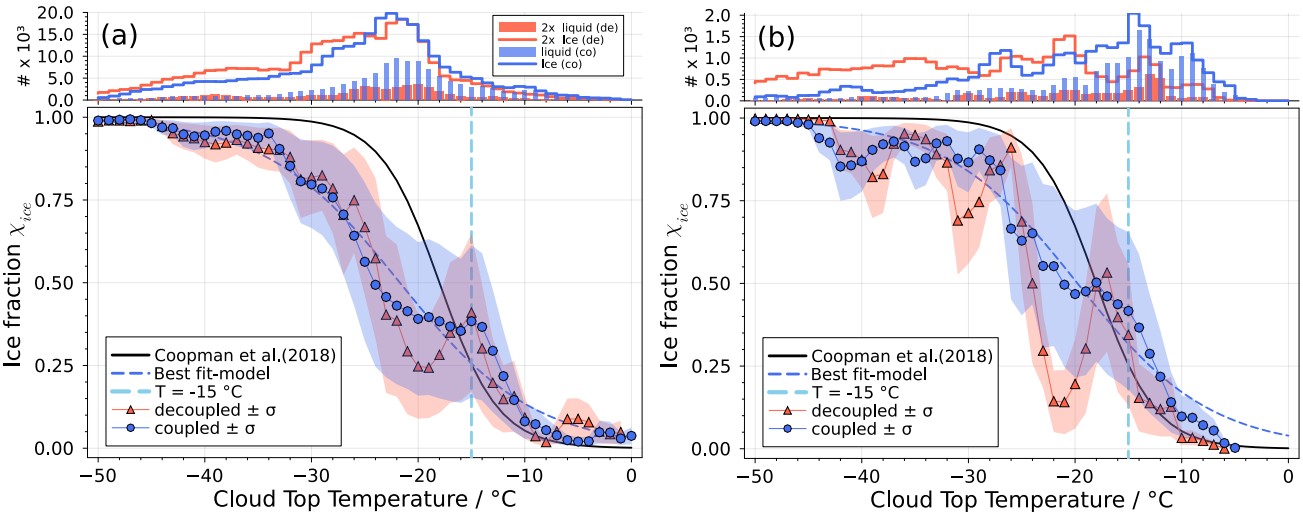

**Figure 10.** Top panels: Histograms of the number of occurrence for liquid (bars) and ice (lines) water path for coupled ("co", blue) and decoupled ("de", orange) cases. For visualization purposes, the decoupled number of occurrence has been scaled by 2 (2x). Bottom panels: Ice water fraction, as defined by Eq. 5, versus cloud top temperature for (a) all cases, and (b) cases where LF>0.02. Coupled (decoupled) cases are depicted as blue circles (orange triangles), shaded areas represent one standard deviation of the ice water fraction sorted within $1\,°C$ temperature bins. The best fit to the coupled data is shown as dashed blue line.

of $-22\,°C$ and $-24\,°C$ for the decoupled and coupled cases, respectively. For $\chi_{ice}$=75% and higher, the coupled and decoupled
$\chi_{ice}$ curves behave similarly towards homogeneous freezing which has been found to occur within the range $-37°C$ to $-40°C$
(Rogers and Yau, 1991; Pruppacher and Klett, 1997).

For the subset of data with LF>0.02 (Fig. 10 (b) ) the $\chi_{ice}$ coupled and decoupled cases reach a first local maximum at
a colder temperature of $-18\,°C$. Note, however, that the number of cases for LF>0.02 are much smaller by a factor of $\sim$6
(Table 3). Furthermore this local maximum at $-18\,°C$ coincides with $\chi_{ice}$=50% which means it reaches the 50% ice water
fraction at a warmer temperature as compared to the Fig. 10 (a). Moreover, it can also be seen that $\chi_{ice}$ for LF>0.02 follows
more closely the empirical model given as $\chi_{ice}(T) = 0.5[1+tanh(-\beta_0(T-\beta_1))]$ by Coopman et al. (2018) based on satellite-
based observations for the Arctic between 2005 and 2010 during March to September. A slight modification to the empirical
model for $\chi_{ice}$ was done in this study so that $\beta_1$ fits the temperature corresponding to $\chi_{ice}$=50%. The decoupled curve shows a
steep drop on ice fraction to a minimum of about 15% at the temperature of $-20\,°C$ (similar situation as observed in Fig. 10 (a)),
followed by an abrupt increase of $\chi_{ice}$ to 95%. The best fit curve for $\chi_{ice}$ of coupled cases has a more monotonically increase
but with a less steep slope as the empirical model (blue-dashed-line curve).

### 4.2.2 Liquid and ice water content as a function of sea ice

With the aim to confirm the relationship between cloud properties with sea ice lead fraction found in Sect. 4.1 the whole
dataset is analyzed and summarized in Fig. 11. The number of occurrence for the whole dataset of observations containing





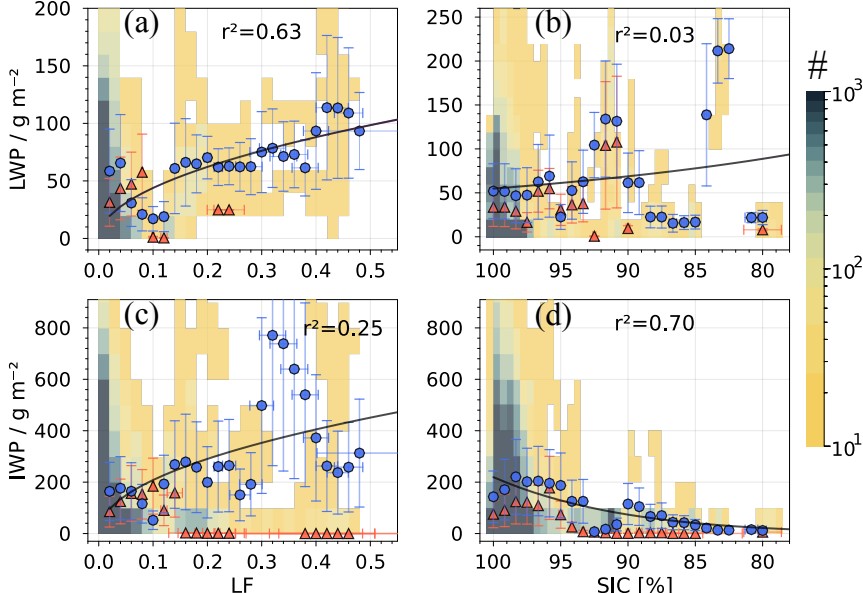

**Figure 11.** Data for the period November 2019-April 2020: Left column, distribution of LWP as a function of observed lead fraction (a) and (c); right column, distribution of IWP as a functions of sea ice concentration (b) and (d). The symbols are the average of the observations within a fixed LF bin-width while the bars indicate 63% of variability within the bin. The colour scale indicated the number of observations within the bins for the whole dataset whereas the symbols represent only the data corresponding to LF>0.02. The coupling status is indicated by orange triangles (decoupled) and blue circles (coupled). The black curves represent the best fit for only the coupled blue circles and the coefficient of determination is given by $r^2$.

either ice or MPC is shown colour-coded in the background (corresponding to the total cloudy sky in Table 3). Overlaid on every subfigure the data for LF>0.02 (see Table 3 for reference on number of observations for LF>0.02) is depicted as binned average for water vapour transport coupled (blue circles) and decoupled (orange triangles) with the corresponding standard deviation indicated by the bars. The best fit curve to the coupled data (blue circles) is indicated by the black curves and the coefficient of determination $r^2$ is given.

The positive correlation between LWP and LF is evident from Fig. 11 (a), with the later being responsible for 63% of the variability observed of LWP as indicated by $r^2$. Albeit an apparent reduction of LWP is found from LF between 0.02 and 0.08, this can be due to the circumstantial lack of LWP observations around LF of 0.1. However, the fit is robust for data with LF from 0.1 and higher. When comparing LWP to sea ice concentration, the positive relation is certainly weak or arguable nonexistent since the analysis shows that SIC can only explain 3% of variability of LWP values (Fig. 11 (b)). This result is strongly influenced by the LWP values less than $40\,\mathrm{g\,m^{-2}}$ paired with SIC between 90% and 80%. When excluding these data points, the correlation of LWP and SIC is enhanced. It is important to remark that due to the different sensors and retrieval methods, SIC and LF are not equivalent or interchangeable, meaning that higher LF observations not necessarily correspond to low SIC as demonstrated in Fig. 3. Therefore even after SIC is adjusted to reduce retrieval underestimations during April





(see Appendic C), SIC can still contain ill-posed SIC retrievals that can attribute LWP values to uncertain SIC, e.g., a range of
LWP values mapped to low SIC.

Regarding the IWP sensitivity to changes in LF and SIC, Fig. 11 bottom row indicates a moderate positive correlation with
LF ($r^2$ of 0.25 for LWP vs. LF). On the contrary, when IWP is related to SIC, the relation is opposite as for the case IWP
versus LF, mainly for the region of SIC between 80 and 96%, with only a fairly increase of LWP when SIC change from 100
to 97%. Important to note that most of the observed high values of IWP are related to deep precipitating cloud systems. When
the data is additionally constrained to cloud depths below 3 km (not shown) the IWP is drastically reduced to values mainly
below $150 \, \mathrm{g \, m^{-2}}$ and the graphs in Fig. 11 bottom row do not show any significant positive correlation with LF, whereas the
LWP versus LF and SIC are only subjet to negligible changes. Concluding, a positive correlation of IWP versus LF is mostly
caused by deep cloud systems.

## 5   Conclusions and Outlook

Based on the methodology developed in Sect. 3 applied to the case study of 18 November, 2019 and to the complete MO-
SAiC wintertime observations (November 2019 - April 2020) for statistical analysis, the following can be concluded:

- The WVT coupled cloud observations outnumber the decoupled cases by at least a factor of 1.6 (61% vs. 39% from all
  cloudy cases in Table 3). However, this factor can range between 10 up to 100 for a given total water path (Fig. D3).
  With the LWP found to be mostly below $150 \, \mathrm{g \, m^{-2}}$) whereas for IWP most of the data points are above $100 \, \mathrm{g \, m^{-2}}$.

- When the LF>0.02, coupled clouds are statistically lower and thicker in height and depth, respectively.

- Coupled clouds have significantly warmer cloud top temperatures, with a vast majority of clouds having a temperature
  inversion at cloud top, thus implying a stratified stable cloud layer.

- The cloud microphysical properties such as droplet effective radii distribution does not shown any clear indication of
  relevant differences between LF≤0.02 and LF>0.02 for WVT coupled clouds. With only a slightly increase of probability
of occurrences of $\bar{\mathrm{r}}_{\mathrm{eff}}$ between 10 and 25 $\mu m$ for decoupled cases. The effective radius for ice does not show a clear
  dependence on LF for coupled cases. For decoupled cases the $\bar{\mathrm{r}}_{\mathrm{eff}}$ are larger for LF>0.02 than for LF<0.02. This suggests
  that the ice $\bar{\mathrm{r}}_{\mathrm{eff}}$ of decoupled clouds seems to be more sensitive to LF which is counter intuitive.

- The distribution of LWP and IWP as a function of sea ice lead fraction and sea ice concentration reveals that only LWP
  and LF are strongly related ($r^2 = 0.63$). For the case of LWP versus SIC, the correlation is weakly negative ($r \sim -0.17$),
with SIC only explaining 3% of the LWP variability. Important to note that SIC<90% was mainly observed during the
  April 2020 (Fig. C1), this MOSAiC period is being extensively studied due to the occurrence of warm air intrusions into
  the central Arctic conducive to inaccuracies on AMSR2 based retrievals (Krumpen et al., 2021; Rückert et al., 2023).
  Given that in April there was not 8% open water reported around RV *Polarstern*, the bias correction to the AMSR2-



MODIS merge product (presented in Appendix C) still has inaccuracies leading to the attribution of LWP at lower SIC
values, hence affecting the correlation between LWP and SIC.

– The occurrence of LF>0.02 is correlated with increases of IWP. This is however contradictory with the result found for
IWP versus SIC, where decreasing SIC is strongly correlated with reduction of IWP. Important to note, however, is that
when cases with cloud depth larger than 3 km are excluded (not shown), the relation between LWP versus LF and SIC
does not differ from the results in Fig. 11. The main difference is the that for these situations no significant relation
between IWP versus LF ($r^2 = 0.0$) is found. For IWP versus SIC, when only cloud depths below 3 km are considered,
the pattern remains similar ($r^2 = 0.33$) when al cloud depths are considered. This is an indicator that sea ice leads have
no trivial effect on IWP but are only strongly correlated to LWP.

– The ice fraction of total water content $\chi_{ice}$ depicts major differences when the entire dataset is compared to cases
with LF>0.02. Mainly in the region of heterogeneous ice formation between $-10$ to $-32\,°$C, with the observations
with LF≤0.02 have a pronounced peak at about $-15\,°$C (temperature where maximum growth of ice crystals takes
place (Rogers and Yau, 1991)). For the subset of data with LF>0.02 the maximum $\chi_{ice}$ peaks is displaced to slightly
colder temperature ($-17\,°$C). The $\chi_{ice}$=50% is reached at warmer temperatures ($-18\,°$C) for LF>0.02 as compared to
the entire dataset ($-22\,°$C). Both cases contrast largely to results reported by Westbrook and Illingworth (2011) for mid-
latitude observation where $\chi_{ice}$=50% was reported to happen at $-27\,°$C based, however, on cloud observations only for
temperatures below $-10\,°$C.

The results found in this study are presented using the lead fraction as constrain to distinguish effects on cloud properties.
Based on SIC data from the MODIS-AMSR2 merged products which has a spatial resolution of 1 km is sufficient to detect
large leads. However, this resolution and merged retrieval product are not sufficient to resolve most small leads. Thus the novel
product of lead fraction (LF) with spacial resolution of 700 m based on SENTINEL-1 SAR satellite divergence data is of
utter importance to prescribe the influence of sea ice leads over the cloud properties. The MOSAiC observations of LWP and
IWP for cloud layers coupled to WVT have been analyzed as a function of LF and depict a clear positive relation between
LWP and IWP with LF for coupled cases. When compared to SIC, LWP has a less pronounced positive relation and IWP
even exhibits a negative correlation. The dataset constrained to LF>0.02 comprises only about 10% of the total data containing
clouds, nevertheless it exhibits significant differences compared to the whole dataset regarding cloud liquid base height, cloud
thickness, cloud top temperature and lapse-rate. In contrast, micro-physical properties like liquid and ice effective radius have
a less pronounced difference based on LF but rather on their coupled status.

Previous studies have already shown differences on various cloud properties when classified by surface coupling or obser-
vations over ocean or sea ice. For instance, Gierens et al. (2020) found, using observation from Ny-Ålesund in Svalbard, that
surface coupled persistent MPC contain about twice as much more liquid as the decoupled clouds. The total amount of con-
densed water was higher for coupled persistent MPC which let them to suggest the existence of a humidity source which is not
available for the decoupled MPC. This suggestion can be confirmed by the present study since the WVT serves as humidity
source from sea ice leads. Papakonstantinou-Presvelou et al. (2022), using satellite products for large-scale clouds below 2 km



in the Arctic, found strong ocean/sea ice contrast of ice crystal number concentration over sea ice than over ocean, with this difference being enhanced for temperatures between 0 and $-10\,°C$ and clouds located south to 70 °N latitude. Although our

study encompasses a different scale, we found that highest values of IWP are concentrated at low LF or high SIC.

Griesche et al. (2021) reported contrasting ice formation in summer Arctic clouds when separated by surface coupling from observations on-board the RV *Polarstern* in 2017. With larger number of ice containing clouds corresponding to surface coupled clouds between $-10\,°C$ and $-5\,°C$ cloud minimum temperature. Although we use ice water fraction instead, our study found contrasting differences for WVT coupled versus decoupled cases, but mainly located in the range between $-15\,°C$ and

$-25\,°C$ cloud top temperature.

Of significant importance is the understanding of the micro-physical processes related to the interaction between water vapour, liquid and ice growth revealed by the ice water fraction which exposed a clear asymmetry when separated by the coupling to WVT. This result requires further investigation since such an impact of WVT on cloud properties has not be reported previously.

The presented study puts into consideration a methodology to study the influence of sea ice to cloud properties based on the observations from the MOSAiC expedition. Although MOSAiC is unprecedented in terms of detailed dataset, it only comprises one winter. Therefore, a similar study is being extended to a period from 2012 to 2022 at the Western Arctic using data from the ARM North Slope of Alaska (NSA) site in Utqiávik, which has a comparable instrumental suite as the RV *Polarstern* during MOSAiC . Moreover, recent improvements for cloud phase classification are being implemented. This refers to Schimmel et al.

(2022) who use radar Doppler spectrum for the detection of liquid layers above lidar attenuation, and thus provides the potential to significantly improve the cloud phase target classification which can then be used to support the findings of this study.

*Code and data availability.* Data were obtained from the atmospheric radiation measurement (ARM) user facility, a U.S. department of energy (DOE) office of science user facility managed by the biological and environmental research program. The merged MODIS-AMSR2 1 km SIC product is publicly available at the University Bremen data repository seaice.uni-bremen.de. HATPRO MWR data is public and

published (Ebell et al., 2022). PollyXT and Cloudnet classification was provided by H. Giesche and is available on request. Lead fraction data based on divergence product from SENTINEL-1 is currently under publication process by L. von Albedyll and can be provided on request. Cloudnet target classification has been calculated using Tukiainen et al. (2020).

**Appendix A: Cloud decoupling criteria**

The virtual potential temperature is defined as:

$$\theta_v = \theta \left( \frac{1 + \frac{q_r}{\epsilon}}{1 + q_r} \right) \tag{A1}$$

where $q_r$ is the water vapour mixing ratio [g g$^{-1}$], $\epsilon \approx 0.622$ is the ratio of dry air to wet air gass constant, and $\theta$ is the potential temperature $\theta = T \left( \frac{P_0}{P} \right)^\kappa$ in K, being $\kappa \approx 0.286$ the ratio of the dry air gas constant and specific heat capacity for constant pressure.





In order to estimate the mixing layer below the cloud, the radiosonde profiles are used first to compute the virtual potential

temperature according to equation A1 and then the cumulative variance of the potential temperature is calculated following:

$$\sigma_\Sigma^2(z) = \sum_i^z \theta_v^2(i) - \left[\sum_i^z \theta_v(i)\right]^2 \forall i = \{\text{CBH}, \dots, 0\} \tag{A2}$$

Equation A2 is evaluated starting at cloud base height (CBH) downwards until surface level, i.e., $i$=0. The mixing layer

below the cloud is then assigned to the altitude where Eq. A2 first surpasses a threshold value of 0.05 $\text{K}^2$, as explained in main

text Sect. 3.3.

Similarly for the cloud driven mixing layer above the cloud top, Eq. A2 is evaluated from cloud top height (CTH) upwards

until a threshold of 0.01 $\text{K}^2$ is fulfilled. A stricter criteria for the upper threshold is due to the fact that there are cases where the

temperature inversion happens inside the cloud and not necessarily above CTH, for these cases $\theta_v$ can already be at a regime

of adiabatic cooling thus its variability might be small. This was also reported by Sedlar et al. (2012) for central Arctic oceans

where they found cloud top was frequently located at 100-200 m above the temperature inversion base.

**Appendix B: Derivation for the gradient of water vapour transport**

The integrated vapour transport (IVT) is defined as the vertical integral of horizontal vapour fluxes (Zhu and Newell, 1998)

and normally used to identify AR whenever either the IVT threshold of 250 $\text{kg s}^{-1}\text{ m}^{-1}$ is exceeded or when the IVT exceeds

the 85th percentile of a climatologically varying value. IVT can be calculated by integrating the module of the wind vector $\boldsymbol{v}_w$

times the specific humidity $q_v$:

$$\text{IVT} = -\frac{10^2}{g} \int_{P_0}^{P_1} |q_v \cdot \boldsymbol{v}_w| \, dP \tag{B1}$$

where $V_w$ is the horizontal wind speed [$\text{m s}^{-1}$], $q_v$ the specific humidity [$\text{g g}^{-1}$], $P$ atmospheric pressure [hPa], and $g$ the

constant of gravity 9.81 $\text{m s}^{-2}$. The factor $10^2$ in Eq. B1 expresses IVT in units of $\text{kg m}^{-1}\text{ s}^{-1}$ after the integration normally

performed from surface reference pressure $P_0$ to a nominal $P_1$=300 hPa.

The vertical gradient of WVF is obtained starting from the definition of IVT given by Eq. B1 and applying the derivative

with respect to the vertical component $z$,

$$\frac{d}{dz}\text{IVT}(P) = -\frac{10^2}{g}\frac{d}{dz}\left[\int_{P_0}^{P} |q_v \cdot \boldsymbol{v}_w| \, dP\right] \tag{B2}$$

by using the chain rule the integration variable can be changed from P to z as,

$$\frac{d}{dz}\text{IVT}(z) = -\frac{10^2}{g}\frac{d}{dz}\left[\int_{z_0}^{z}\left(|q_v \cdot \boldsymbol{v}_w|\frac{dP}{dz}\right)dz\right] \tag{B3}$$



thus the integral can be canceled by the outer derivative resulting in:

$$\frac{d}{dz}\mathrm{IVT}(z) = -\frac{10^2}{g}\left( |q_v \cdot \boldsymbol{v}_w| \frac{dP}{dz} \right) \qquad (B4)$$

hereafter we rename the derivative of IVT with respect to the z variable in Eq. B4 as gradient along the vertical for the water vapour transport as $\nabla_z\mathrm{WVT}(z)$ which has units of $\mathrm{kg\ s^{-1}\ m^{-2}}$, resulting in

$$\nabla_z\mathrm{WVT} = -\frac{10^2}{g}\,|q_v \cdot \boldsymbol{v}_w|\,\frac{dP}{dz} \qquad (B5)$$

Note that we started from the definition of IVT given by Eq. B1 where the horizontal wind speed is used. Some authors
prefer to define IVT by means of the wind zonal and meridional components $U$ and $V$, respectively. It is important to mention that both definitions are not mathematically identical, thus producing slightly different results.

## Appendix C: Sea ice concentration offset correction

As reported first by Krumpen et al. (2021) and extensively studied by Rückert et al. (2023), there were several events of warm air mass intrusion (WAI) during the MOSAiC expedition mainly in spring 2020. Those WAI events have fostered inaccuracies
in some sea ice concentration retrievals (Fig. 1 and Fig. 9 by Rückert et al. (2023) and Krumpen et al. (2021), respectively). This is particularly the case for products from algorithms that use MWR polarization information at 36 GHz, i.e. the ASI algorithm. In the context of the present study those inaccuracies have fostered a misclassification of cloud properties when sorted as a function of SIC due to the SIC offset. Therefore it is paramount to correct the SIC product when the offset due to WAI is present. This section describes the details of this correction.

The underestimation by MODIS-AMSR2 SIC retrievals can be observed from Fig. C1 middle-panel. For the period of mid-February to end of May 2020 considerable disagreement has been found in the SIC products obtained by the MODIS-AMSR2 and the OSI-SAF retrievals. Unfortunately, the resolution of the OSI-SAF product is 25 km which is not enough to resolve small leads relevant to this study. Therefore, OSI-SAF is only being used as reference product and we do not imply OSI-SAF as error-free retrieval or for absolute values of SIC.


In Fig. D2 can be seen the advantage to detect leads by the MODIS-AMSR2 product (a) as compared to the OSI-SAF (b). Conversely OSI-SAF sea ice concentration has the advantage of not being affected by the WAI events due to its different retrieval algorithm, which has been corroborated by Krumpen et al. (2021) and Rückert et al. (2023). Therefore those two products are being used in order to fix the SIC bias using OSI-SAF while still keeping the high resolution variability of the
MODIS-AMSR2 product.

The time series of the OSI-SAF and merged MODIS-AMSR2 products averaged over the 50 km radius around RV *Polarstern* is shown in Fig. C1. Clearly confirms the WAI compromised sea ice concentration mainly from February to May 2020 as reported by Rückert et al. (2023).

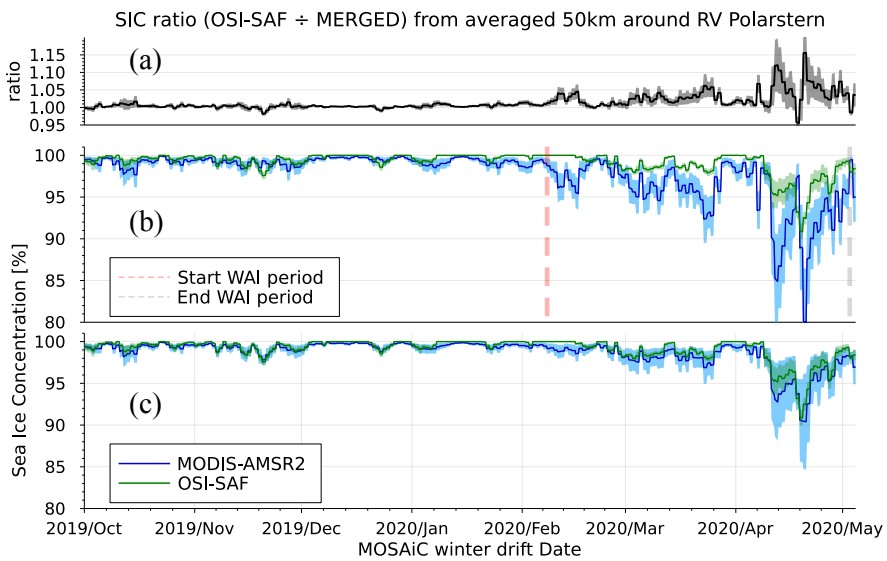

**Figure C1.** Frop top to bottom: (a) the ration of OSI-SAF by MODIS-AMSR2 SIC products, close to 1 indicates that both products are in average similar within the sector of study; (b) SIC from MODIS-AMSR2 (blue) and from OSI-SAF product (green) averaged within 50 km around RV *Polarstern*. The initial and end of reported warm air intrusions are marked by the dashed red and grey vertical lines; (c) same as above but with the MODIS-AMSR2 SIC corrected (blue). In all panels the shared areas correspond to 1 standard deviation.

In order to have both products statistically comparable, the MODIS-AMSR2 SIC is averaged by assuming a truncated normal distribution with 0 and 100 as lower and upper distribution limits. The averaging was performed within a 10 km grid centered at every OSI-SAF coordinate grid thus a MODIS-AMSR2 and OSISAF dataset is achieved with the same spatial resolution of the OSI-SAF grid. Then the ratio of those two products is calculated as an indicator of over or under estimation of ASI relative to OSI-SAF. The ratio is shown in Fig. C1 (top panel) for the entire period of interest. It can be seen that the greatest discrepancy occurs from middle February to May (in agreement with findings by Rückert et al. (2023)) with the OSI-SAF SIC retrieval being up to 15% higher than the MODIS-AMSR2. On the other hand the ratio rarely reaches ±2% for lower or higher MODIS-AMSR2 SIC outside the WAI period (mainly winter 2019) corroborating the fact that when no WAI events are experienced both product are comparable.

Since both SIC products are provided at a 24 hours temporal resolution the SIC ratio has been estimated on a daily bases. Every MODIS-AMSR2 SIC section of interest around RV *Polarstern* is then corrected by the SIC-ratio to match in average the OSI-SAF but keeping the high-spatial resolution and the variability within this sector of study. The comparison of this procedure is depicted in Fig. C1 bottom panel where the offset free SIC is overlapped to the OSI-SAF as visual assessment for the feasibility of the method.

**Appendix D:  Supporting Material**



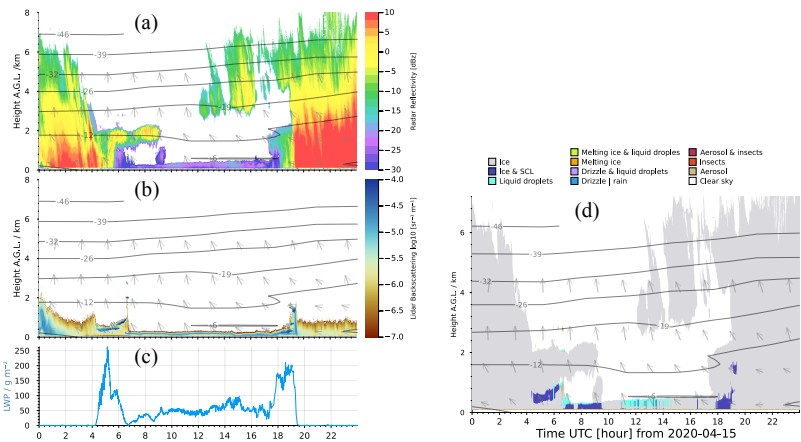

**Figure D1.** Synergy of remote sensing instruments on board of the RV *Polarstern* for 15 April 2020. (a) KAZR cloud radar reflectivity factor; (b) PolyXT lidar backscattering coefficient; (c) liquid water path from the microwave radiometer. The Iso-therms and wind vectors are obtained from the weather model at selected altitudes and time steps. (d) Cloudnet classification.

## D1 Case Study for 15 April, 2020

Sea ice observations for 15 April, 2020. In this example, both MODIS-AMSR2 (1 km resolution) and SENTINEL-1 (700 m) detected the lead, where as OSI-SAF (25 km resolution) is not able to detect the lead.

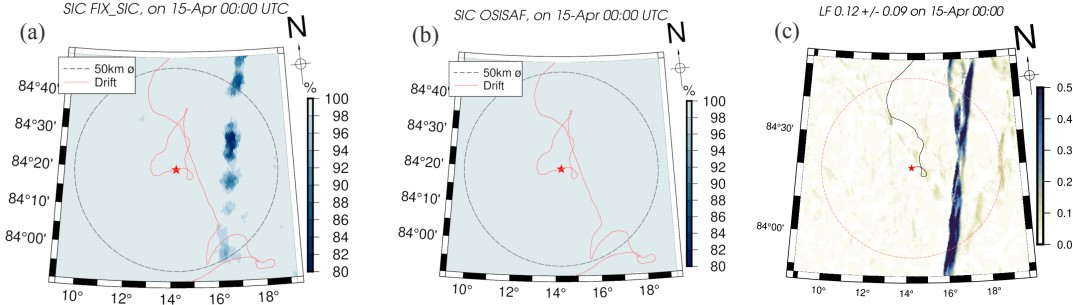

**Figure D2.** Satellite-based sea ice observations for 15 April, 2020. (a) SIC from MODIS-AMSR2 merge product after being offset corrected (see appendix C). (b) SIC from OSI-SAF. (c) LF from SAR SENTINEL-1. The images are centered at the position of the RV *Polarstern* (red star) at the given date. The RV drift is indicated by the red-line, the dashed circle indicates a the 50 km radius region of interest.

## D2 Statistical Analysis Additional Results

In Fig. D3 (a) is shown that when the frequency of occurrence is binned within intervals of total water path, the coupled cases are approximately 10 times more frequent than decoupled cases. Fig. D3 (a) also shows the distribution of coupled and
decoupled cases with respect to cloud phase and integrated water path (colour scale).





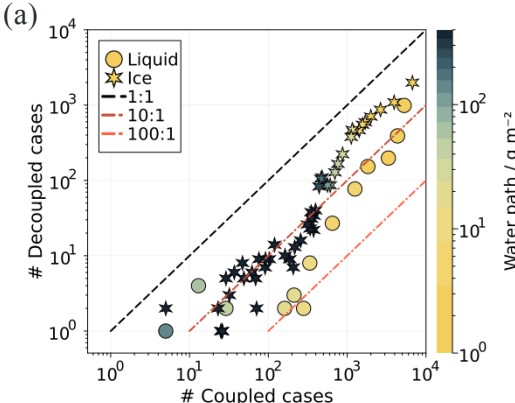
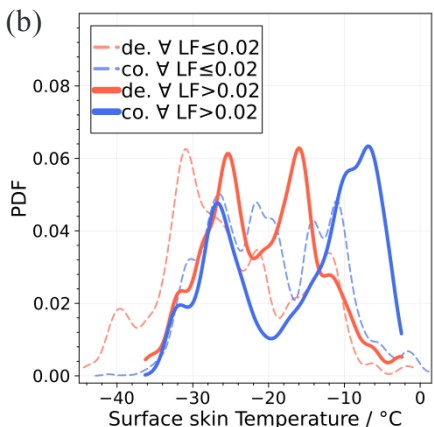

**Figure D3.** (a) the distribution of cloud observations as a function of the integrated water path for coupled versus decoupled cases. The colour scale indicates the cloud water path for ice (stars) and liquid (circles). (b) PDF for the surface skin temperature from the GNDIRT sensor.

It can be seen that liquid clouds are mostly found below 200 g m$^{-2}$ and are more frequent than the ice clouds. Finally the PDFs for surface skin temperature are presented in Fig. D3 (b), showing a asymmetric coupled/decoupled bimodal distributions. Cases with LF>0.02 are associated with a considerable warm surface for coupled cases with one peak located at $-6\,°$C, whereas the decoupled PDF shows one peak at $-17\,°$C.

**D3   List of Acronyms and variables**

| Acronym | Description |
| --- | --- |
| KAZR | Ka Zenith Radar |
| MWR | Microwave Radiometer |
| CEIL10m | Ceilometer 10 m resolution |
| INTERPSONDE | Interpolated Radiosonde |
| RV | Research Vessel |
| CO | RV *Polarstern* Central Observatory |
| AMSR2 | Advanced Microwave Scanning Radiometer 2 |
| MODIS | Moderate Resolution Imaging Spectroradiometer |
| MPC | Mixed-phase Clouds |
| ASI | Artist sea ice algorithm |
| OSI SAF | Ocean and Sea Ice Satellite Application Facility |
| SAR | Synthetic Aperture Radar |



| Symbol | Description | Unit |
|---|---|---|
| LWP | liquid water path | $\mathrm{g\ m^{-2}}$ |
| IWV | integrated water vapour | $\mathrm{kg\ m^{-2}}$ |
| LWC | liquid water content | $\mathrm{g\ m^{-3}}$ |
| IWC | ice water content | $\mathrm{g\ m^{-3}}$ |
| IWP | ice water path | $\mathrm{g\ m^{-2}}$ |
| IVT | integrated water vapour transport | $\mathrm{kg\ s^{-1}\ m^{-1}}$ |
| $\nabla_z$WVT | vertical gradient of water vapour transport | $\mathrm{kg\ s^{-1}\ m^{-2}}$ |
| CBH | cloud base height | m |
| CTH | cloud top height | m |
| CMLH | cloud mixing-layer height | m |
| LF | sea ice lead fraction | - |
| SIC | sea ice concentration | % |
| Ze | radar equivalent reflectivity factor | dBz |
| $V_D$ | mean Doppler velocity | $\mathrm{m\ s^{-1}}$ |
| $SW_D$ | Doppler spectral width | $\mathrm{m\ s^{-1}}$ |
| $\beta$ | lidar attenuated backscattering coefficient | $\mathrm{m^{-1}\ sr^{-1}}$ |
| $\delta$ | lidar depolarization ratio | - |
| $P$ | atmospheric pressure | Pa |
| $q_v$ | specific humidity | $\mathrm{g\ g^{-1}}$ |
| $W_s$ | horizontal wind speed | $\mathrm{m\ s^{-1}}$ |
| $W_d$ | wind direction from North | ° |
| $T_v$ | virtual temperature | K |
| $\theta_v$ | virtual potential temperature | K |
| $\Sigma\sigma_i^2(\cdot)$ | cumulative variance function | $K^2$ |
| $Ri_b$ | bulk Richardson number | - |
| $T_{gnd}$ | skin surface temperature | K |
| $\Gamma_{\mathrm{cloud}}$ | cloud layer temperature lapse rate | $°\mathrm{C\ km^{-1}}$ |
| $\chi_{ice}$ | ice water fraction in cloud layer | - |
| $\bar{\mathrm{r}}_{\mathrm{eff}}$ | cloud layer average effective radius | $\mu\mathrm{m}$ |



*Author contributions.* PSG performed the data analysis and led the manuscript writing. HKL as project leader contributed to the research design and result analysis. LvA produced the sea ice divergence and lead fraction from SAR SENTINEL-1 data. HG provided Cloudnet and the low level cloud flag product for MOSAiC . GS provided the sea ice concentration products. All co-authors contributed with discussion and revision of the manuscript .

*Competing interests.* The authors declare no conflict of interest.

*Acknowledgements.* We thank all those who contributed to the Multidisciplinary drifting Observatory for the Study of Arctic Climate (MO-SAiC) Consortium, and made this drift observatory possible! (Nixdorf et al., 2021). Our gratitude to the DOE ARM program for openly provide the data from its Mobile Facility 1 on board of RV *Polarstern* . To the open Sea Ice data repository from the University of Bremen and the EUMESAT Ocean and Sea Ice Satellite Application Facility (OSI SAF). This work was supported by the *Deutsche Forschungsgemeinschaft* (DFG) funded Transregio-project TR-172 "Arctic Amplification $(AC)^3$" (grant 268020496). Thanks to Nils Hutter who developed the directional filter used to remove noise from the SAR divergence fields.



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
