# Peer review of "Asymmetries in cloud microphysical properties ascribed to sea ice leads via water vapour transport in the central Arctic"

_EGUsphere, 2023_

## Referee Comment (RC1)

**Overview**

Sea ice leads plays a central role in surface energy balance and affect the overlying atmosphere over polar regions via the highly efficient exchange in both heat and moisture. Using observations from the MOSAiC filed campaign and novel vapor transport-based method, the authors investigate the influence of upstream sea ice leads on downstream cloud properties in the wintertime Arctic and observe different (asymmetric) micro- and macro-physical cloud properties when leads present versus not present. I appreciate the authors integrate a pile of data to make this happen and find the results overall reasonable and valuable, which offers detailed insights to understand the sea ice-cloud interactions. I do believe this work can be published on ACP after revisions listed below.

**Major comments**

1. On coupled vs decoupled, lead vs sea ice: By reading the title only, I would expect the authors are referring this asymmetry to cloud properties observed with and without upwind sea ice leads. Yet, the abstract and the main results are instead focusing on cloud comparisons under coupled versus decoupled scenarios. So I wonder are the authors trying to emphasize the asymmetries of cloud property differences between coupled and decoupled cases when lead fraction is small (<0.02) versus large (>0.02)? It might be helpful to clarify this in the title and main text in the first place.

2. If the above-stated is the case, a follow-up question emerges. I see the value to sample the cases based on surface coupling state, and most often clouds are coupled with the surface when leads are present (also evidenced by Fig.8). There might be abnormal cases (e.g., when clouds are surface-coupled even with the absence of leads and vice versa), but I would expect these should rarely happen. Based on Table 3, it seems that the authors do detect such cases and the surface-coupled cases are in fact quite often (up to 64%) when lead fraction is less than 0.02, which I would take it as sea ice scenario considering (a) the uncertainty in the divergence-based lead fraction product and (b) the focused area (i.e., a conical sector centered at Polarstern and extended up to 50 km radius and angular span of 5 degrees) is relatively small and so is the actual lead area. In other words, I am worried about the reliability of the method used to detect coupled case when lead fraction is quite small (for example less than 0.02 in this study). This is somewhat exhibited by looking at the example case (Fig. 6c): the maximum $\nabla$WVT (~9 g/m2/s) detected near the surface is not very distinguishable as there is a second maximum (~8 g/m2/s) right above it at ~ 1.5 km high. In addition, how far the detected leads relative to the cloud observation site might be another factor influencing the surface-coupling state detection. With that said, I wonder can the authors provide some convincing evidence to demonstrate the coupled case when leads are almost absent and explain why? If the above-stated (i.e., emphasize the asymmetries of cloud property differences between coupled and decoupled cases when lead fraction is small (<0.02) versus large (>0.02)) is not the case, given the uncertainty in coupling state detection, I don't see the necessity to divide the samples into coupled versus decoupled cases when lead fraction < 0.02, such

as the results in Figure 9. Figures like 8 or 11 showing the entire range of lead fraction tell a nice story.

**Minor comments**

1.  L10-11. "cloud-driven layer extending above the cloud top and below the cloud base, respectively"
2.  L11-13: These are very detailed information on data, should be put in Method or elsewhere instead of Abstract.
3.  L16: The comparison between coupled and decoupled clouds are not clear. Readers might think the decoupled clouds are also low-level clouds but only thinner than that coupled ones. Please rephrase it.
4.  L73-74: Sect. 4, 4.1 and Sect. 4, 4.2 are misleading. Suggestion either using Section 4.1 and Section 4.2 or just merger them into one. Also, fully spell all "Sect." in the text to avoid confusion with other short names.
5.  L147: provide -> provided
6.  L168-169: Spell out "CO" and other places appropriate; there are already too many acronyms which reduce the readability of the paper. You want the readers to remember the most important ones, like LF. Plus, "co" also represents coupled in the paper.
7.  L175: "," -> "."
8.  L180: are of having- > are having
9.  Table 2: Table caption should appear above the associated table.
10. L269: downwind -> upwind
11. L273: of below cloud base's -> or below cloud base's
12. L277: to be take place-> to take place
13. L310: Fig. 5 -> Fig.7
14. L327: missed a comma before relationships
15. L328-331: These details on method should be better to put in the caption instead of the main text to make the manuscript more concise and readable.
16. Legend in Fig.8: the circle and triangle filled with color is unnecessary and misleading. Suggest use unfilled ones.
17. L351: Grammarly incorrect sentence. Please rephrase.
18. L387: any evidence for the argument that temperature inversions are found above the cloud base for those coupled case?
19. L395: any girds with mixed-phase clouds? How these are considered in the calculation of Eq(5).
20. L405: why choose based on temperature range?
21. L455: Discussion, Besides summarizing and listing these observed results and comparing to previous studies, one should say more about what these information can infer and provide insights for the community. More discussion regarding this would benefit the readers.

---

## Referee Comment (RC2)

**Review of Egusphere-2023-623**

This manuscript presents the investigation of the relationship between sea ice lead fractions and cloud micro- and macro-physics during the MOSAiC field campaign. The study is constructed with an introductory case study followed by statistical analysis. The statistical analyses show that the coupled cases are under the influence of enhanced water vapor transport from the leads area, hence the enhanced moisture supplies contribute to the cloud properties. I found the manuscript to be well constructed and logical in the narrative. Nevertheless, I do have a few comments and suggestions listed below, which should be considered and addressed before potential publication.

**General Comment.**

The statistical results seem to be based on the available cloudy samples regardless of the cloud types. At least, the cloud type criteria are not clearly stated in the manuscript (i.e., in Fig. 1 and D1, there are already two types of cloud systems: stratiform and convective). I am concerned that the intrinsic differences in the microphysical processes of those different cloud systems would impair or blur the robustness of the results, especially in the interpretation of the comparisons between coupled/decoupled cases and different LF circumstances (i.e., the discussions regarding Fig. 9 to Fig. 11). For instance, the differences in the LWP and IWP between coupling and LF categories could potentially be more influenced by the cloud thicknesses.

I wonder if you have considered enhancing the robustness of the analysis in a more controlled environment, e.g., confining the cloud selection to stratiform or convective clouds only. Please give it some thought.

**Minor Comment.**

L86. Please define HATPRO.

L109. 'Advanced Microwave Scanning Radiometer 2 (AMSR2)'

L184. Can you provide the precisions or the estimated errors for the Cloudnet retrievals, preferably, compared with the aircraft in-situ measurements?

L264. According to Appendix A, do you mean 0.05 $K^2$ here for estimating the sub-cloud mixing layer right?

L377. It seems that the liquid and ice effective radii shown here range from non-precipitating to heavy-precipitating clouds, have you considered the aerosols (e.g., sea salts) advected along the WVT pathway that served as CCN or INP and affect the cloud microphysics, and in turn, bias the results?

L424. If, in the case of LF > 0.02, presumably implied in the aforementioned discussion, it indicates more moisture supply to the cloud layer. How do you interpret the difference in the $\chi_{ice}$ dips ($\sim$ -20°$C$) of the decoupled cases, i.e., any ascribable relations between the increased moisture supply and the heterogeneous freezing process? Similar questions can be asked for the dips in $\sim$ -30°$C$ and -40°$C$.

L442. Since it is mentioned here that the SIC and LF are not equivalent, it would be interesting to show if there is any relationship between SIC and LF, i.e., a scatter plot of conical SIC vs. LF.

L447. '…for IWP vs. LF'

L448. Do you mean 'with only a fairly increase of IWP when SIC change from 100 to 97%'?

L559. 'WVT'

L614. In Table 3 the ratio of coupled to decoupled is ~6:4, while here states that the coupled cases are 10 times more frequent than the decoupled cases when binned by water path. Can you clarify?

Figure 11. The first sentence of the caption conflicts with the subfigures. LWP plots should be (a) and (b), while IWP plots should be (c) and (d). And can you clarify why the bars are sometimes discrete within the same LF bin?

---

## Author Comment (AC1)

**REVIEWER No. 1**

Major comments

1.On coupled vs decoupled, lead vs sea ice: By reading the title only, I would expect the authors are referring this asymmetry to cloud properties observed with and without upwind sea ice leads. Yet, the abstract and the main results are instead focusing on cloud comparisons under coupled versus decoupled scenarios. So I wonder are the authors trying to emphasize the asymmetries of cloud property differences between coupled and decoupled cases when lead fraction is small (<0.02) versus large (>0.02)? It might be helpful to clarify this in the title and main text in the first place.

*Reply: Thank you for the observations. We want to emphasize the fact that cloud observations above the RV Polarstern cannot directly be associated to sea ice leads observed afar from the RV Polarstern, but rather by means of using a mechanism to link these two observables. This mechanism is proposed in the manuscript to be the water vapour transport (WVT). We do agree that the title does not reflect this idea, therefore we changed the title to "Asymmetries in cloud microphysical properties ascribed to sea ice leads via water vapour transport in the central Arctic".*

2. If the above-stated is the case, a follow-up question emerges. I see the value to sample the cases based on surface coupling state, and most often clouds are coupled with the surface when leads are present (also evidenced by Fig.8). There might be abnormal cases (e.g., when clouds are surface-coupled even with the absence of leads and vice versa), but I would expect these should rarely happen. Based on Table 3, it seems that the authors do detect such cases and the surface-coupled cases are in fact quite often (up to 64%) when lead fraction is less than 0.02, which I would take it as sea ice scenario considering (a) the uncertainty in the divergence-based lead fraction product and (b) the focused area (i.e., a conical sector centered at RV *Polarstern* and extended up to 50 km radius and angular span of 5 degrees) is relatively small and so is the actual lead area. In other words, I am worried about the reliability of the method used to detect coupled case when lead fraction is quite small (for example less than 0.02 in this study). This is somewhat exhibited by looking at the example case (Fig. 6c): the maximum $\nabla$WVT (~9 g/m2/s) detected near the surface is not very distinguishable as there is a second maximum (~8 g/m2/s) right above it at ~ 1.5 km high. In addition, how far the detected leads relative to the cloud observation site might be another factor influencing the surface-coupling state detection.

With that said, I wonder can the authors provide some convincing evidence to demonstrate the coupled case when leads are almost absent and explain why? If the above-stated (i.e., emphasize the asymmetries of cloud property differences between coupled and decoupled cases when lead fraction is small (<0.02) versus large (>0.02)) is not the case, given the uncertainty in coupling state detection, I don't see the necessity to divide the samples into coupled versus decoupled cases when lead fraction < 0.02, such as the results in Figure 9. Figures like 8 or 11 showing the entire range of lead fraction tell a nice story.

**Reply**: We generally refer to cloud coupling to the WVT which is related to the presence of upwind sea ice leads, in other words, the coupling concept used in the manuscript encompass the sea ice lead-WVT-cloud system. We did that because sea ice leads are not generally

present just below the observed cloud and because about only 6.5% of cases show a cloud surface coupling in the classical sense, this is due to the persistent intermittent boundary layer near surface level which serves as decoupling agent between the surface and the cloud above (as illustrated in Fig. 4).

It is true that coupling case happens even though no sea ice leads are observed, this is not generally abnormal since it happens due to: 1. WVT exist and it is either weak or comes from a location further away than 50 km, 2. There is in fact a sea ice lead but it is outside our considered range i.e. 50km radius centered at RV *Polarstern*. The latter implies that clouds will be classified as coupled (because is interacting with WVT) but will also be classified to cases with LF<0.02. That is why the reviewer noticed (point (b) in the comment) based on the Table 3. Regarding the reviewer comment point (a), in fact the methods used in the study to detect sea ice leads have limitations and it can produce under- or over-estimation of sea ice openings. We found the Sentinel 1A divergence product for LF the best product for this study because of its higher resolution and its ability to only detect leads when they open, avoiding therefore the consideration of newly frozen leads which has been argued to serve as a dissipation mechanism for low level clouds (Li et al., 2020). Furthermore, as explained in the methodology, we intentionally use the vertical gradient of the WVT to find its maximum and hence the wind direction at that altitude. This is done in order to stress the WVT at lower altitudes under the hypothesis that this WVT is more likely to be interacted with leads within the 50km range. WVT maximums located at higher altitudes can happen and can be weaker in magnitude, this still can be due to leads further away from the 50km, but clouds above RV *Polarstern* are either classified as decoupled (because WVT maximum is at higher altitude) or classified as coupled but with LF<0.02. This is why we separate the statistical analysis between coupled/decoupled cases and LF<0.02 and LF>0.02. Unfortunately, with the current available observations, we cannot determine how far the WVT filament extends, so that only WVT originated within the 50km can be considered. Back-trajectories analysis show that during MOSAiC the air masses have origins far from the Artic circle (Silber & Shupe, 2022) that is why in our manuscript we state that the WVT does not exclusively originate from the sea ice leads but rather that the sea ice lead release of latent and sensible heat interact with the WVT and this serves as conveyor belt to transport the energy (and eventually sea spray as aerosol sources from leads) toward the cloud observed above RV *Polarstern*.

Furthermore, we separate the coupling state when LF<0.02 to have an insight on the situation where WVT is present and leads can be located at ranges further than 50km. For instance, when the probability distribution function (PDF) - of a certain cloud property - shows basically the similar shape for coupled cases with LF<0.02 and LF>0.02, it means there might be still leads located further away which produce similar PDF e.g. same PDF maximum location but less frequent. On the contrary, for coupled cases and with LF>0.02, the PDFs are different from the decoupled ones (e.g. multiple peaks versus mono-modal distributions) is an indication that the leads-WVT-cloud coupling system is separating the observations into two distinguishable distributions.

**Minor comments**

1. L10-11. "cloud-driven layer extending above the cloud top and below the cloud base, respectively"

**Reply**: Thank you, the abstract has been simplified and corrected.

2. L11-13: These are very detailed information on data, should be put in Method or elsewhere instead of Abstract.

Reply: We do agree with the reviewer observation. The abstract has been simplified.

3. L16: The comparison between coupled and decoupled clouds are not clear. Readers might think the decoupled clouds are also low-level clouds but only thinner than that coupled ones. Please rephrase it.
*Reply: It has been rephrased as "Clouds coupled to WVT are found to have mostly lower cloud base and larger thickness than decoupled clouds".*

4. L73-74: Sect. 4, 4.1 and Sect. 4, 4.2 are misleading. Suggestion either using Section 4.1 and Section 4.2 or just merger them into one. Also, fully spell all "Sect." in the text to avoid confusion with other short names.
*Reply: Thank you for the suggestion. We changed to Section although Sect. seems to be the requirement of the journal specifications.*

5. L147: provide -> provided
*Reply: This has been corrected, Thanks.*

6. L168-169: Spell out "CO" and other places appropriate; there are already too many acronyms which reduce the readability of the paper. You want the readers to remember the most important ones, like LF. Plus, "co" also represents coupled in the paper.
*Reply: We do agree with the observation by the reviewer, we did take out the acronym CO from whole manuscript.*

7. L175: "," -> "."
*Reply: Corrected.*

8. L180: are of having- > are having
*Reply: This has been corrected.*

9. Table 2: Table caption should appear above the associated table.
*Reply: The caption of all tables have been placed above.*

10. L269: downwind -> upwind
*Reply: Thank you for noticing this mistake, it has been corrected.*

11. L273: of below cloud base's -> or below cloud base's

*Reply: Thank you, it has been corrected.*

12. L277: to be take place-> to take place
*Reply: Thank you, it has also been corrected.*

13. L310: Fig. 5 -> Fig.7

*Reply: Thank you, it has been corrected.*

14. L327: missed a comma before relationships

*Reply: Thank you, it has been corrected.*

15. L328-331: These details on method should be better to put in the caption instead of the main text to make the manuscript more concise and readable.
*Reply: The sentence has been simplified and the caption of Fig. 8 is better described.*

16. Legend in Fig.8: the circle and triangle filled with color is unnecessary and misleading. Suggest use unfilled ones.
*Reply: We apologize for the confusion. The intention was to also show the relationship with temperature. Now in Fig. 8 the color coded symbols have been removed, but for consistency purposes with the whole manuscript we keep the symbols colored as blue-circles and orange-triangles for  for coupled and decoupled cases, respectively.*

17. L351: Grammarly incorrect sentence. Please rephrase.
*Reply: Thanks for point out this mistake, the grammar has been reviewed and the sentence corrected: "The 18 November, 2019 case study encompasses a situation where the observed clouds have a well-defined correlation with LF situation up-wind, mainly due to the occurrence of a single cloud layer.".*

18. L387: any evidence for the argument that temperature inversions are found above the cloud base for those coupled case?
*Reply: Temperature inversions above cloud base have also been reported by (Sedlar et al., 2012) where they found the inversion not only can take place at cloud top height but also inside the cloud. This is something we want to analyze better with our dataset, and determine whether there is any pattern on the thermodynamic structure of the cloud layer when separated by cloud coupling. However this is not within the scope of the present manuscript.*

19. L395: any girds with mixed-phase clouds? How these are considered in the calculation of Eq(5).
*Reply: This is a good observation. Indeed, one reason for why we use the ice water fraction, instead of  the commonly used  fraction of number of pixels with mixed-phase clouds to the total number of cloud pixels,  is to consider the amount of water within the mixed-phase clouds into the calculation of Eq. (5). When a grid cell is classified as mixed-phase by the Cloudnet algorithm, then that grid cell has a value for ice and liquid water content. This is integrated along the cloud layer and therefore included in Eq. (5). If in our study we were to consider ice cloud occurrence instead of water content, then mixed phase clouds would only contribute to the denominator of Eq. (5).*

20. L405: why choose based on temperature range?
*Reply: The reason is to be consistent with most of the scientific literature where cloud top temperature at ranges from -40 to 0°C is used. This way we can also compare our findings with the ones from other studies.*

21. L455: Discussion, Besides summarizing and listing these observed results and comparing to previous studies, one should say more about what these information can infer and provide insights for the community. More discussion regarding this would benefit the readers.
*Reply: Thanks for the comment. In Discussion section after the summarizing the findings a paragraph has added to stress this point:  "The findings presented here can be used as valuable constrains to evaluate cloud microphysical parameterizations for the Arctic system models. Since sea ice leads are not explicitly resolved in such models, lead-averaged surface heat flux, and its influence on clouds, is of considerable interest for the parameterization of energy exchange* (Gryschka et al., 2023)*. The different features of ice water fraction $\chi_{ice}$ , as a function of cloud top temperature, found for coupled and decoupled cloud cases are a result*

*that deserves to be deeply investigated by validating it with long-term observations but also by a better understanding  the modelling of cloud microphysics that can lead to explain the findings."*

**Bibliography**

Gryschka, M., Gryanik, V. M., Lüpkes, C., Mostafa, Z., Sühring, M., Witha, B., & Raasch, S. (2023). Turbulent Heat Exchange Over Polar Leads Revisited: A Large Eddy Simulation Study. Journal of Geophysical Research: Atmospheres, 128(12), e2022JD038236. https://doi.org/10.1029/2022JD038236

Li, X., Krueger, S. K., Strong, C., Mace, G. G., & Benson, S. (2020). Midwinter Arctic Leads Form and Dissipate Low Clouds. Nature Communications, 11(1), 206. https://doi.org/10.1038/s41467-019-14074-5

Sedlar, J., Shupe, M. D., & Tjernström, M. (2012). On the Relationship between Thermodynamic Structure and Cloud Top, and Its Climate Significance in the Arctic. Journal of Climate, 25(7), 2374–2393. https://doi.org/10.1175/JCLI-D-11-00186.1

Silber, I., & Shupe, M. D. (2022). Insights on sources and formation mechanisms of liquid-bearing clouds over MOSAiC examined from a Lagrangian framework. Elementa: Science of the Anthropocene, 10(1), 000071. https://doi.org/10.1525/elementa.2021.000071

---

## Author Comment (AC2)

**REVIEWER No. 2**

This manuscript presents the investigation of the relationship between sea ice lead fractions and cloud micro- and macro-physics during the MOSAiC field campaign. The study is constructed with an introductory case study followed by statistical analysis. The statistical analyses show that the coupled cases are under the influence of enhanced water vapor transport from the leads area, hence the enhanced moisture supplies contribute to the cloud properties. I found the manuscript to be well constructed and logical in the narrative. Nevertheless, I do have a few comments and suggestions listed below, which should be considered and addressed before potential publication.

**General Comment.**

The statistical results seem to be based on the available cloudy samples regardless of the cloud

types. At least, the cloud type criteria are not clearly stated in the manuscript (i.e., in Fig. 1 and

D1, there are already two types of cloud systems: stratiform and convective). I am concerned that the intrinsic differences in the microphysical processes of those different cloud systems would impair or blur the robustness of the results, especially in the interpretation of the comparisons between coupled/decoupled cases and different LF circumstances (i.e., the discussions regarding Fig. 9 to Fig. 11). For instance, the differences in the LWP and IWP between coupling and LF categories could potentially be more influenced by the cloud thicknesses.

*Reply: We understand the concern. For our analysis, in case of observations with multiple cloud layers, we always consider the  single cloud layer  which is closest to the maximum of WVT.  It happens, however, that for coupled cases that layer is commonly the lowest layer and for decoupled clouds can be located higher up, that is the reason why we do not specifically consider only stratiform low level clouds. By doing so, some decoupled clouds could have been dismissed and we would have only analyzed coupled low level clouds but the idea of the methodology is to have a reference to compare the coupled clouds, i.e. using the decoupled clouds as a reference of cases without interaction with the sea ice leads. It is correct that cases as shown in D1 includes convective systems even though they are still considered one cloud layer, but we found that by constraining the cloud thickness below 2.5km, the relationship between LWP vs LF and SIC practically is preserved, whereas the relationship between IWP vs LF and SIC is considerable reduced. This is an indication that the presence of sea ice leads mainly have an effect on the liquid fraction of the mixed-phase clouds e.g. liquid water content, liquid cloud base height.*

[Figure]

**Figure 1 Same as in the manuscript Figure 11 but the symbols are the result of constraining the cloud top height to cases below 2.5km to represent low level clouds.**

I wonder if you have considered enhancing the robustness of the analysis in a more controlled environment, e.g., confining the cloud selection to stratiform or convective clouds only. Please give it some thought.

*Reply: Thanks for the comment. This has been considered by carefully selecting the case study from November 18th, 2019, to explain the methodology. In this case study two cloud systems were present, first, a low stratiform cloud and later a convective deep precipitating cloud (Figure 1 and 5 in manuscript). To highlight the effect introduced to our statistical analysis, we reproduced the manuscripts's Figure 11 by only considering cloud top heights below 2.5km (Figure 1 in this document). The* Figure 1 *reveals the robust positive relationship between LWP and LF ($r^2$ =0.58, versus $r^2$=0.63 for all cloud heights in manuscript Figure 11) is originated mainly from the low level clouds. Regarding IWP, it can be seen that when cloud top height is constrained to 2.5km clouds have values for IWP below 100 g $m^{-2}$ , with the coupled cases (blue circles) having systematically larger IWC  than the decoupled cases (orange triangles). Therefore, we can conclude that the main source of higher ice water content are deep precipitating systems rather than stratiform low level clouds.*

**Minor Comment.**

L86. Please define HATPRO.

*Reply: HATPRO stands for Humidity And Temperature PROfiler and it has been defined in Line 86.*

L109. 'Advanced Microwave Scanning Radiometer 2 (AMSR2)'

*Reply: Thank you. It has been corrected*.

L184. Can you provide the precisions or the estimated errors for the Cloudnet retrievals, referably, compared with the aircraft in-situ measurements?

*Reply: Cloudnet retrieves ice water content (IWC) based on a relationship to reflectivity and temperature (Hogan et al., 2006). Comparison with in-situ aircraft measurements shown a root mean square error in retrieving IWC of around +50% to -33% for the temperature ranges from -20°C to -10°C. Whereas for temperatures below -40°C the error rises to +100% to -50%. As stressed by (Hogan et al., 2006) these uncertainties are, however, large due to the small sample volume of the aircraft probes. The main source of uncertainty comes from the radar reflectivity factor $Ze$. To have an insight into the effect of $Ze$ uncertainty into the total ice water path (IWP), we performed an estimation of the relative error for the IWP when the reflectivity is changed by ±3, ±10 and ±20% of the original measured value. This results in changes on retrieved IWP as shown in Figure 2 for the manuscript's case study from November 18th 2019. It can be seen that by modifying the reflectivity by $Ze$ +20% the relative error obtained in IWP has a practically constant value of -12%, while for $Ze$ -20%, IWP is slightly lower than 13% of the original retrieved value. For the cases of ±10% and ±3% the retrieved IWP lays within a constant margins of ±6.5% and ±1.9%, respectively, regardless of the absolute value for IWP. Even though Figure 2 confirms the sensitivity of IWP on the reflectivity factor, it is realistic to assume that the uncertainty of radar reflectivity is within the ±3% of measured value, which gives a solid ±2% of IWP uncertainty*.

[Figure]

**Figure 2 Top panel: November 18th, 2019 Cloudnet IWP retrieved from measured reflectivity with grey shadow area indicating the range of IWP retrieved when the reflectivity is modified within ±20%. Bottom panel, left y-axis: the IWP residuals (original - modified) for cases when Ze is modified by ±3 (green shading), ±10 (orange shading), and ±20% (blue shading). Bottom panel, right y-axis: relative IWP error.**

*Regarding the liquid water content (LWC), Cloudnet first classify the grids containing liquid droplets and estimates the LWC profile based on the theoretical adiabatic LWC gradient from cloud base. The adiabatic LWC is then scaled so that its integral matches the microwave radiometer measurement of liquid water path (LWP). Therefore the main source of uncertainty is due to the LWP retrieved from the microwave radiometer. A multi-frequency radiometer as HATPRO using a quadratic regression to retrieve LWP can have a root mean square error of about 15.4 g m⁻² (Hogan et al., 2006; Löhnert & Crewell, 2003). Based on a similar sensitivity exercise as for IWP, in we found that the retrieval error is covered by a variation of less than 20% on the total LWP to be scaled by the adiabatic LWC approach. That gives us a confidence that the LWP used in this work is within a 20% uncertainty.*

[Figure]

**Figure 3 Same as Figure 2 but for the LWP.**

L264. According to Appendix A, do you mean 0.05 K2 here for estimating the sub-cloud mixing layer right?

*Reply: The text is correct, we used 0.01 K² for the estimation of the sub-cloud mixing layer.*

L377. It seems that the liquid and ice effective radii shown here range from non-precipitating to heavy-precipitating clouds, have you considered the aerosols (e.g., sea salts) advected along the WVT pathway that served as CCN or INP and affect the cloud microphysics, and in turn, bias the results?

*Reply: We agree with the reviewer that this is certainty the case. Nevertheless there is neither direct nor remote sensing measurements of advected aerosols along the WVT path. During MOSAiC expedition aerosols and INP have been sampled at the RV Polarstern location (Creamean et al., 2022), showing that INP concentration are found to be persistent among the months from October to April mainly between the range of temperatures from -25°C to -15°C*

*(Creamean et al., 2022) , with large INP sampling during periods with high lead occurrence and wind speeds above 5 m s⁻¹. Therefore , as highlighted by (Creamean et al., 2022) the high fractional occurrence of ice in clouds below 3km (low-level clouds) in winter implies that observed small INPs could serve as important role in cloud ice formation. However due to the fact that the surface is predominantly frozen the local source of INPs is locally limited. Thus, it is plausible to support the hypothesis that leads play an important role as sources of sea spray by windy conditions during the wintertime. We consider that the sea ice leads as sources of INP like sea-spray can be advected along the WVT, and therefore included in our analysis as part of the coupled/decoupled classification. However, since no continues INP sampling has been performed we cannot separate our dataset based on INP concentration, but we agree that such type of analysis is an important source of information to narrow down the leads effects on cloud properties.*

L424. If, in the case of LF > 0.02, presumably implied in the aforementioned discussion, it indicates more moisture supply to the cloud layer. How do you interpret the difference in the χice dips (~ -20°C) of the decoupled cases, i.e., any ascribable relations between the increased moisture supply and the heterogeneous freezing process? Similar questions can be asked for the dips in ~ -30°C and -40°C.

*Reply: This is an important concern about the finding presented in the manuscript. The reviewer is right by interpreting that for LF>0.02 more moisture supply should be present, however for the decoupled case it means that the cloud layer is not interacting with the moisture supply. Our finding on the $\chi_{ice}$ asymmetry between coupled and decoupled, e.g. dips at -20°C, -30°C and -38°C, is certainly surprising. The dips on $\chi_{ice}$ for decoupled cases can be interpreted as a consequence of the cloud layer not being supplied by any moisture or nucleating particles which can be originated from leads. On the contrary the $\chi_{ice}$ coupled cases (blue symbols) indicates a continuous source of moisture and INPs from leads, therefore the mixed-phase clouds are likely unstable the Wegener-Bergeron-Findeisen process favors the ice growth at the expense of vapour deposition, the heterogeneous freezing process*. (Danker et al., 2022) using CloudSat-Calipso DARDAR product for clouds below 2.5km, have also reported the increase on occurrence of mixed-phase and decrease of super cooled liquid clouds at temperatures around -15°C, although they only consider cloud top temperatures up to -20°C. For similar *dips on $\chi_{ice}$* at lower temperatures no other references have been found.

L442. Since it is mentioned here that the SIC and LF are not equivalent, it would be interesting to show if there is any relationship between SIC and LF, i.e., a scatter plot of conical SIC vs. LF.

*Reply: We mentioned that SIC and LF are not equivalent with the intention to stress that those are independent sea ice states estimates, with LF being more suitable to resolve small divergent leads (between two consecutive satellite overpasses) with a nominal resolution of 700 m. Whereas SIC is a merged product from MODIS (thermal, 1km resolution) and AMSR2 (microwave, 3.124 km resolution) more suitable to detect larger leads or partially frozen leads. In other words, a simple conversion of variable like SIC = 100\*(1 – LF) is not applicable for our case. In Figure 4 is the scatter plot of the conical sector within 6° around the wind direction and centered at the location of the RV Polarstern.*

[Figure]

**Figure 4** Sea ice concentration (SIC) vs lead fraction (LF) from November 2019 to April 2020 as scatter plot for the median (left panel) and the inter quantile region (IRQ) of the SIC vs. LF distribution within the conical sector centered at the RV Polarstern (right panel).

L447. '...for IWP vs. LF'

*Reply: Thank you for the correction.*

L448. Do you mean 'with only a fairly increase of IWP when SIC change from 100 to 97%'?

*Reply: Yes, that is what we meant. Thank you for the observation, it has been corrected.*

L559. 'WVT'

*Reply: Thanks, it has been corrected to WVT.*

L614. In Table 3 the ratio of coupled to decoupled is ~6:4, while here states that the coupled cases are 10 times more frequent than the decoupled cases when binned by water path. Can you clarify?

*Reply: Table 3 indicated the total cases. Figure D3 (a), separated the number of cases within intervals of total water path, most of the points with total water path higher than 100 g m⁻² (dark green symbols) lay along the 10:1 red dashed-line, i.e. coupled cases occur about 10 times more often than decouple cases.*

Figure 11. The first sentence of the caption conflicts with the subfigures. LWP plots should be (a) and (b), while IWP plots should be (c) and (d). And can you clarify why the bars are sometimes discrete within the same LF bin?

*Reply: Thank you for pointing out the mistake. It is corrected to: "Top row, distribution of LWP as a function of observed LF (a) and SIC (b); bottom row, distribution of IWP as a functions of LF (c) and SIC (d)".*

**Bibliography**

Creamean, J. M., Barry, K., Hill, T. C. J., Hume, C., DeMott, P. J., Shupe, M. D., et al. (2022). Annual cycle observations of aerosols capable of ice formation in central Arctic clouds. Nature Communications, 13(1), 3537. https://doi.org/10.1038/s41467-022-31182-x

Danker, J., Sourdeval, O., McCoy, I. L., Wood, R., & Possner, A. (2022). Exploring relations between cloud morphology, cloud phase, and cloud radiative properties in Southern Ocean's stratocumulus clouds. Atmospheric Chemistry and Physics, 22(15), 10247–10265. https://doi.org/10.5194/acp-22-10247-2022

Hogan, R. J., Mittermaier, M. P., & Illingworth, A. J. (2006). The Retrieval of Ice Water Content

from Radar Reflectivity Factor and Temperature and Its Use in Evaluating a Mesoscale Model. Journal of Applied Meteorology and Climatology, 45(2), 301–317. https://doi.org/10.1175/JAM2340.1

Löhnert, U., & Crewell, S. (2003). Accuracy of cloud liquid water path from ground-based microwave radiometry 1. Dependency on cloud model statistics. Radio Science, 38(3). https://doi.org/10.1029/2002RS002654

---

## Author Comment (AC3)

**REVIEWER No. 3**

The article "Asymmetries in winter cloud microphysical properties ascribed to sea ice leads in the central Arctic" studies the impact of the presence of leads on cloud properties based on data from the MOSAiC campaign. The authors efficiently use the synergy of different datasets to constrain several parameters (coupled vs. decoupled cases, different lead fractions...). The study highlights the importance of considering the leads to study the properties of Arctic clouds. The dataset, the method and the analyses seem to be robust and the results are convincing. The authors show some interesting results: For example, the lead fraction has an important influence on the cloud thickness and on the ice water path. I have a few comments (see below) that I would like the authors to consider, but the topic and content of the paper are within the scope of the journal, so I recommend publication.

**General comments:**

1. I wonder if the authors looked at the effect of melt ponds or what they think about it. Would it have a similar effect as the leads (maybe weaker effect)?

   *Reply: As reported by (Creamean et al., 2022), melt ponds occurrences coincide with high concentration of INPs sampled during the MOSAiC expedition. Therefore melt ponds can be thought as sources of nucleating particles necessary for cloud formation. However, as also shown by (Creamean et al., 2022) in their Figure 2, the occurrence of melt ponds happens during the Arctic summer mainly after May, when air temperature is close to zero or even slightly above 0°C, which makes melt ponds not very efficient as sources of sensible heat. On the contrary sea ice leads are efficient sources of latent and sensible heat during winter where air temperatures ranged from -45°C to -10°C. This is one reason our manuscript focuses in the wintertime, so the effects be leads can be stressed and better isolated from the cloud observations. This remarks are added to the manuscript introduction in lines 57 to 64.*

2. The authors use the Cloudnet dataset based on observations to retrieve cloud properties. Like any observation, Cloudnet should have an error in the observations and in the retrievals, but this is not shown here. I expect that this uncertainty should appear in the results, or at least be discussed.

   *Reply: This is an important observation and a column in Table 2 has been added with the uncertainties reported by the literature where retrievals are based on. Moreover, a sensitivity study has been performed for cases where the radar reflectivity is modified by ±3, ±10, and ±20% of the original values and found that the final ice water path (IWP) relative error ranges from ±1.9% to ±13%. See answer to reviewer 2 for detailed explanation.*

3. Methodology: The considered wind profile is measured at the RV Polarstern and not at the leads. Therefore, I think Figure 4 is misleading, but I understand that it is considered constant between the leads and RV Polarstern: How true is this hypothesis, have the authors quantified the possible biases from a change in WVT along the way (between the lead and Polarstern)?

   *Reply: Thank you for pointing out this, the manuscript's Figure 4 has been modified to indicate the wind profile is measured at the RV Polarstern. In order to assess the hypothesis that the wind direction at maximum WVT gradient can be assumed constant*

*within the 50km radius considered in the manuscript, we performed back-trajectory analysis using the Lagrangian back-trajectory tool Lagranto (Sprenger & Wernli, 2015) . The trajectory of WVT is tracked from the altitude where the wind direction is taken i.e. maximum gradient of WVT, this shows that the assumption is quite plausible since the back-trajectories show a considerable agreement with the assumed wind direction within the 50km radius. In Figure 1 is shown the back-trajectory for WVT (gray band) at different times. Only between 18:00 UTC and 20:00 UTC it has been observed that the back-trajectory moves earlier towards west as compared with the wind direction showing a deviation between the conical sector (black lines) and the back-trajectory, which can be explained by inaccuracies on the hourly ERA5 input for Lagranto. The animation showing the back-trajectories for the entire day is added as supplementary material*.

[Figure]

**Figure 1 Sea ice lead fraction from November 18th, 2019. The two radial back lines encompass the conical sector considered within 50km radius for the analysis based on the wind direction. The gray band indicated the WVT back-trajectory path using Lagranto, with the circles along the path indicating hourly backward steps of the trajectories.**

4. Do the authors look at what types of clouds we are mainly looking at in the study? In terms of mixed-phase clouds, are they mainly the typical mixed-phase clouds with precipitating ice below a liquid layer, or are they more mixed?

   *Reply: Mixed-phase clouds are the dominant cloud type during Arctic winter and MOSAiC. In the analysis we are considering non-precipitating mixed-phase clouds as well as clouds with precipitating ice below the liquid layer as it is shown in the case study from November 18th, 2019. We are starting to do further research whether or not the amount of precipitating ice below the liquid layer has any correlation with the presence of*

*sea ice leads, but for the current manuscript precipitating mixed-phase clouds are not separated from the analysis.*

5. Figure 7: From the plot, it appears that clouds are mixed phase in the coupled situations and only ice in the decoupled situations. Is this always the case? Can the authors comment on this? And the next question is: Could we use the presence of mixed phase to detect coupled situations (or vice versa)? Could we use the presence of leads (and coupled situations) to detect mixed phase clouds (or vice versa)?

   *Reply: Thanks for the question. The case shown in the manuscript's Figure 7 can be misleading in the sense that mixed-phase clouds can also be classified as decoupled since the classification is based on the location of WVT relative to the cloud layer. As shown in manuscript's Figure 9 (a) and (d), decoupled mixed-phase clouds are also present in the analysis. Regarding the second question: Although our results show that coupled mixed-phase clouds are more frequent, it is not feasible to identify those mixed-phase clouds directly as coupled since there are still a considerable amount of decoupled mixed-phase clouds with and without lead fraction. Based in our statistical analysis it can be infered that coupled mixed-phase clouds occur with a 63% frequency when LF>0.02, but we cannot detect the mixed-phase clouds only based on the presence of leads.*

6. I was wondering if the authors considered the effect of aerosols. Leads could be a source of marine aerosols and therefore affect the thermodynamic phase of the cloud. I guess Polarstern might have measurements of this. The increase in moisture would be the most important effect on cloud properties, but aerosols might not be negligible.

   *Reply: Aerosols in fact have been observed during MOSAiC. In particular samples of ice nucleating particles (INP) have been collected during MOSAiC. (Creamean et al., 2022), showing that INP concentration are found to be persistent among the months from Octover to April mainly between the range of temperatures from -25°C to -15°C (Creamean et al., 2022) , with higher INP amount sampled during periods with high lead occurrence and wind speeds above 5 m s$^{-1}$. Therefore , as highlighted by (Creamean et al., 2022) the high fractional occurrence of ice in clouds below 3km (low-level clouds) in winter implies that observed small INPs could serve as important role in cloud ice formation. However due to the fact that the surface is predominantly frozen the local source of INPs is locally limited. Thus, it is plausible to support the hypothesis that leads play an important role as sources of sea spray by windy conditions during the wintertime. We consider that the sea ice leads as sources of INP like sea-spray can be advected along the WVT, and therefore included in our analysis as part of the coupled/decoupled classification. However, since no continues INP sampling has been performed, we cannot separate our dataset based on INP concentration.*

**Minor comments:**

1. Title: I do not like the term "Asymmetries" because it emphasizes more a difference than an asymmetry. Also, I found that this term is more associated with geographical differences, but that may be just me, but I recommend changing the title.

   *Reply: Thank you for the comment. We use the term Asymmetries to emphasizes the differences found between coupled and decoupled clouds at different sea ice lead fraction states. Therefore it is highlighted that the methodology for coupling the sea ice leads with the clouds via the WVT mechanism, proposed in the manuscript, has*

*uncovered asymmetries in the statistical distributions of coupled and decoupled clouds. In other words, if the methodology presented were able to produce symmetric coupled/decoupled distributions then the conclusion would be that sea ice leads coupled to clouds via WVT have no effect on the cloud properties*.

2.  Abstract: There is a lot of technical details in the abstract that could be removed here.

    *Reply: Thank you for the suggestion, the abstract has been simplified*.

3.  Line 42: We usually refer to the Wegener-Bergeron-Findeisen process. Then the citation Wegener 1911 could be added
    *Reply: Thank you for notice that. We refer now as the Wegener-Bergeron-Findeisen process and the citation has been updated*.

4.  Wegener, A. (1911). Thermodynamik der atmosphäre. JA Barth
    *Reply: Thank you for the information, the citation has been updated*.

5.  Line 118: "We note that leads…" Do the authors mean that in this situation the leads are considered to be sea ice? If so, I wonder if they quantify the error from this.
    *Reply: This has been reported by (Rückert et al., 2023) during warm air intrusions (WAI) in April 2020 where thin ice hampers some sea ice concentration retrieval methods. One reason to use the merged MODIS-AMSR2 SIC product is to circumvent this problem since the MODIS sensor is sensitive to thermal emission thus not affected by the thin ice events. The estimate error can be observed in the manuscript's appendix Figure C1 where the April WAI events overestimate sea ice openings up to 15%. The Sentinel 1A lead fraction product is based on the divergence/convergence of the sea ice, therefore not sensitive to thin ice but rather to the relative movement of the sea ice*.

    Line 160: "described as following", at first, I thought the authors were explaining the identification of sea ice leads, but they are describing the potential influence of leads on cloud properties. I suggest changing the sentence.
    *Reply: The identification of sea ice leads has been extensively explained by (von Albedyll et al., 2021) and it is not within the scope of the manuscript. In line 160 we were describing the methodology used to link the sea ice leads with the clouds via the water vapour transport. The sentence has been simplified and adapted to highlight this (now Line 165)* "The conceptual model proposed to identify the influence of sea ice leads on the cloud properties observed aloft the RV Polarstern 's central observatory  is depicted in Fig. 4 and described as following:"

6.  Line 166: The acronym CO is confusing because it is already used for coupled. Perhaps it is not necessary to have an acronym for Central Observatory since it is not used that often.
    *Reply: Thank you for the suggestion, CO has now been dismissed as acronym in the manuscript*.

7.  l. 180: are of having -> are having
    *Reply: Thanks you for the correction, it has been amended*.

8.  l.190: "meaning the lidar signal is attenuated by low-level liquid clouds", I am not sure I understand. Does this mean that the algorithm does not detect the low level cloud, but the signal is attenuated by the cloud and therefore the measurements are biased? Have

the authors quantified the effect on the results?

*Reply: This means the lidar is mostly attenuated by detecting the lowest liquid cloud. Cloudnet will classify this lowest cloud as pure liquid or mixed-phase cloud, but in case a second layer of mixed-phase cloud is present above then Cloudnet is not able to classify it as mixed-phase since no information from the lidar is available due to the signal attenuation. This is a well-known limitation of classification algorithms based on radar-lidar synergy. For our analysis we consider only one single cloud layer, and based on our statistical results we found that coupled clouds are mostly low-level clouds therefore less likely to be affected by the limitation of the Cloudnet classification due to lidar signal attenuation. Moreover, a recent algorithm development by Schimmel et al., 2022) has demonstrated that this issue can be improved by exploiting the information from cloud radar Doppler spectra. Application of this technique to the MOSAiC data however requires further data analysis and will be considered for future work, which was indicated at the end of the Discussion and Outlook section, now in Lines 524-526 in the updated manuscript.*

9. l. 201: The constant g appears in equation 1, so it should be defined there.
   *Reply: Thanks for noticing this, constant g is now described after equation 1.*

10. l. 273: CMLH of below -> CMLH or below
    *Reply: Thanks for pointing out the error, this has been corrected.*

11. l. 277: to be take -> to be
    *Reply: Thanks for pointing out the error, this has been corrected.*

12. l. 332: "clear", I would be careful with the term "clear" because some points are not within 3 sigma. Also from Figure 8, I wonder how the fit is done.
    *Reply: Thank you for the comment, "clear" has been deleted. The fit is done by using a power relationship between LWP and LF, i.e. $LWP = a*LF^b$ , with a and b fitting constant. This has only been done to highlight the relationship found within variables and not implying a physical law that link these two properties. Same for the fit in manuscript's Figure 11.*

13. l. 347: "Figure 8 … -10° C km-1" I am not sure I understand the sentence, can the authors rephrase it?
    *Reply: We apologize for the confuse phrasing. The sentence has been simplified and rephrased as "Figure 8 (d) indicates that $\Gamma_{cloud}$ is often close to the moist adiabatic lapse-rate 6.5 °C km$^{-1}$ (dashed horizontal line in Fig. 8(d) ). The negative $\Gamma_{cloud}$ values represent cases with a temperature increase within cloud layer or inversion at cloud top".*

14. Figure 11 caption: the subscripts (c) and (d) are not correct.
    *Reply: Thank you for noticing this. It has been corrected.*

15. Section 5 Conclusion and Outlook: What is missing is a discussion section where the various results are summarized. This is done in Figure 4, but I would go a bit further before the conclusions. I do not think much is needed, but just highlighting what the results bring to the model would be enough.
    *Reply: A paragraph has been added after the list of conclusion points at the Discussion section: "The findings presented here can be used as valuable constrains to evaluate cloud microphysical parameterizations for the Arctic system. The different features on ice*

*water fraction $\chi_{ice}$ found when coupled and decoupled cases as a function of cloud top temperature are analyzed is a result that deserves to be deeply investigated by validating it with long-term observations but also by a better understanding of the model's cloud physic that can lead to explain the finding."*

16. l. 559: WVF -> Do you mean WVT?
    Reply: *Thank you for notice this. It has been corrected*.

**Bibliography**

von Albedyll, L., Haas, C., & Dierking, W. (2021). Linking sea ice deformation to ice thickness redistribution using high-resolution satellite and airborne observations. The Cryosphere, 15(5), 2167–2186. https://doi.org/10.5194/tc-15-2167-2021

Creamean, J. M., Barry, K., Hill, T. C. J., Hume, C., DeMott, P. J., Shupe, M. D., et al. (2022). Annual cycle observations of aerosols capable of ice formation in central Arctic clouds. Nature Communications, 13(1), 3537. https://doi.org/10.1038/s41467-022-31182-x

Rückert, J. E., Rostosky, P., Huntemann, M., Clemens-Sewall, D., Ebell, K., Kaleschke, L., et al. (2023). Sea ice concentration satellite retrievals influenced by surface changes due to warm air intrusions: A case study from the MOSAiC expedition. Retrieved from https://eartharxiv.org/repository/view/5195/

Schimmel, W., Kalesse-Los, H., Maahn, M., Vogl, T., Foth, A., Garfias, P. S., & Seifert, P. (2022). Identifying cloud droplets beyond lidar attenuation from vertically pointing cloud radar observations using artificial neural networks. Atmospheric Measurement Techniques, 15(18), 5343–5366. https://doi.org/10.5194/amt-15-5343-2022

Sprenger, M., & Wernli, H. (2015). The LAGRANTO Lagrangian analysis tool – version 2.0. Geoscientific Model Development, 8(8), 2569–2586. https://doi.org/10.5194/gmd-8-2569-2015

---

## Author Response (AR2)

REVIEWER No. 1

I thank the authors for considering my comments and I am satisfied with the responses, the related changes in the manuscript, and the references provided. I think the study analyzes important aspects that have not been reported before. I found only a few typographical errors: I only have spotted few typos:

*Reply*: *We thank the reviewer for the comments and the typographical errors listed below, they are corrected accordingly.*

- line 154: " One limitation on the lead detection by the divergence-based method is that only detects new openings" -> "is that it only detects"

*Reply*: *Corrected in track-changed document line 149.*

- line 198: "addition to the target classification, The presented case study of 18" -> "classification. The presented"

*Reply*: *Corrected in track-changed document line 201.*

- line 212: "Atmospheric Rivers (AR); (Martin-Ralph et al., 2020)" -> "Atmospheric Rivers (AR) (Martin-Ralph et al., 2020)"

*Reply*: *Corrected in track-changed document line 214.*

- lines 294 and 297: I recommend "Coupled: " instead of "Coupled."

*Reply*: *Thank you for the recommendation, it has been changed in both cases in track-changed document  lines 295 and 297.*

- line 340 "low-lever MPC" -> "low-level MPC"

*Reply*: *The typo is corrected in track-changed document line 339.*

- line 502 ") when al cloud depths a" -> ") when all cloud depths a"

*Reply*: *The mistake has been rectified in track-changed document  line 508.*

- line 563 "n cloud properties has not be reported pre" -> "n cloud properties has not been reported pre"

*Reply*: *Thanks for spotting the error, this has been rectified. In track-changed document line 572.*

REVIEWER No. 2

The authors have addressed my questions well. I would suggest publication with a minor as below.

For one of the major comments -- significance of the coupling cases when LF < 0.02, in addition to the explanation in the response document, the authors really should have made corresponding modifications in the main text. At the minimum level, the last paragraph on page

2 can be incorporated where appropriate. This helps clarify and provide context to better deliver the content.

**Reply**: *We thank the reviewer for the suggestion. In section "4.2 Statistical Analysis" it has been included the explanation for including coupling and decoupling cases when LF<0.02 as part of the description of the analysis presented in that section. Lines 372 to 379 in track-changed document.*

REVIEWER No. 3

Dear Authors,

You have clarified my concern adequately. However, I found that some of the responses are worth mentioning in the main text, as listed below.

**Reply**: *We agree with the reviewers and the concerns have been addressed below:*

Is it better to integrate more of the response to my general comments? For example, in the track-changed manuscript L536-540, the author may want to state that 'the relationship between LWP vs. LF and SIC is practically preserved' when constraining the cloud-top to 2.5km, as well as adding a discussion on it. This would add to the robustness of the result, regardless of the specific cloud type, but more on the coupled vs. decoupled distinction.

**Reply**: *Thank you for the suggestion, the findings regarding cloud-top below 2.5km have been elaborated and included in the results as well as in the conclusion sections. See lines 471 to 480 and 545 to 549 in the track-changed document.*

Similarly, I found that the discussion of IWC/LWC retrieval uncertainties (sources) is informative in assessing the relative error. It is at least worth a few words or sentences in the revised Section 3.1 (L200 - 209).

**Reply**: *we agree that the discussion of retrieval uncertainties is informative, and it has been added to the manuscript in the methodology section when the Cloudnet algorithm and retrievals are described. In the track-changed document in lines 185 to 193.*